# Efficient Multivariate Temporal Point Processes via Monotone Alternating Splines

## Abstract

Temporal point processes (TPPs) have widespread applications across various domains. Compared to modeling the conditional intensity of a TPP, modeling its conditional cumulative intensity function (CCIF) improves efficiency and eliminates approximation error. However, currently, the parametrization of CCIF remains limited and scarce. This paper proposes a novel method called the Monotone Alternating Spline (MAS). By leveraging two distinct interpolation and extrapolation components, MAS provides a broad framework for modeling CCIF, offering greater flexibility and efficiency. Theoretically, MAS's interpolation grants a strong fitting ability, while its extrapolation guarantees robust generalization. Extensive experiments show that MAS achieves superior performance on both synthetic and real-world datasets.

## 1 Introduction

Event sequences are prevalent across various domains, such as criminology (Mohler et al., 2011; Zhou et al., 2020), social networks (Chen & Tan, 2018; Meng et al., 2024), finance (Bacry et al., 2015; Hawkes, 2018), seismology (Ogata, 1998; 1999), and neuroscience (Linderman, 2016; Zhou et al., 2022). These sequences are typically characterized by irregular timestamps in continuous time, which sets them apart from conventional regular time series data. Understanding the dynamics within these sequences provides valuable insights for forecasting future events. Temporal point processes (TPPs), a class of stochastic process models, offer a statistical framework for modeling such event sequences.

The key to characterizing a TPP is the next timestamp's distribution given its history. One of the most common approaches to parametrize TPP is to specify a conditional intensity function (CIF). Numerous methods (Du et al., 2016; Mei & Eisner, 2017; Zuo et al., 2020) leverage deep learning approaches to model the CIF and recover the model parameters via maximum likelihood estimation (MLE). A key drawback of CIF-based modeling, however, is that the TPP likelihood involves integrating the CIF over time. Numerical integration inevitably introduces approximation errors and can significantly degrade computational efficiency. To address this issue, Omi et al. (2019) proposed modeling the cumulative conditional intensity function (CCIF) of a TPP rather than directly modeling CIF. The CIF can be recovered by differentiating CCIF, which yields exact values through automatic differentiation (Baydin et al., 2018). Consequently, CCIF-based modeling eliminates approximation errors and improves computation efficiency.

Despite the advantages of CCIF modeling, parameterizations of CCIF remain scarce. Most existing approaches simply leverage monotone neural networks (MNNs), typically a positive-weighted multilayer perceptron (MLP), to guarantee the monotonicity of the CCIF (Liu, 2024; Omi et al., 2019). These designs suffer from limited architectural flexibility and often overlook other desirable properties of the CCIF. In this paper, we introduce a novel class of CCIF parameterizations—**Monotone Alternating Splines (MAS)**. Compared to MNNs, MAS offers greater generality, flexibility, and efficiency, and enjoys strong theoretical guarantees for both fitting and generalization. Generally speaking, MAS consists of two components: (i) MAS interpolation, which leverages any monotone spline to improve fitting performance; (ii) MAS extrapolation, which employs any simple monotonic function, such as exponential or linear, to ensure global monotonicity. As a general CCIF-modeling framework, MAS is compatible with a wide range of common monotonic splines and extrapolation functions—and can be viewed as an extension of many classic point processes. Moreover, because

MAS admits an explicit derivative, its differentiation no longer depends on automatic differentiation, yielding even better efficiency than MNNs. Theoretically, we prove that MAS's interpolation enhances its expressive power, while its extrapolation guarantees strong generalization. Given sufficient data, MAS recovers the globally optimal solution for any underlying TPP distribution. Meanwhile, experiments on synthetic and real-world datasets further demonstrate MAS's clear advantages.

Our contributions can be summarized as follows:

**(1)** We introduce a novel class of CCIF parameterizations, MAS. Compared to MNNs, MAS satisfies all the requirements for CCIF and offers greater generality, flexibility, and efficiency.

**(2)** The fitting and generalization performance of MAS are theoretically guaranteed. MAS exhibits fitting ability comparable to that of any other TPP parameterization while approximating the globally optimal solution for any ground-truth TPP.

**(3)** Extensive comparisons between MAS and baselines are conducted on both synthetic and real-world datasets. Results show that MAS consistently outperforms all baselines across multiple metrics. Comprehensive ablation studies analyze the contributions of each component in the model.

## 2 PRELIMINARIES

In this section, we provide the background on TPPs, CCIF, and monotone splines. For clarity, all notations used in this paper are listed in Table 5 in Section B.

### 2.1 TEMPORAL POINT PROCESSES

A TPP (Daley & Vere-Jones, 2003) is a probabilistic model that describes sequences of events occurring within a time window $[0, T]$. A realization of a multivariate TPPs is represented as an event sequence $S = \{(t_n, k_n)\}_{n=1}^{N}$, where $N$ is a random variable denoting the number of events. Here, $0 < t_1 < \cdots < t_N < T$ are the event timestamps, and $k_n \in (1, \cdots, K)$ is the mark associated with the $n$-th event. TPPs are characterized by the CIF $\lambda_k(t \mid \mathcal{H}_{t-})$, which defines the instantaneous occurrence rate of a type-$k$ event given the history $\mathcal{H}_{t-} = \{(t_n, k_n) : t_n < t\}$. For convenience, we denote it as $\lambda_k^*(t)$, where the asterisk indicates conditioning on the history up to but not including $t$:

$$\lambda_k^*(t) = \lim_{\delta_t \to 0} \frac{p(\text{type-}k \text{ event} \in [t, t + \delta_t] \mid \mathcal{H}_{t-})}{\delta_t}.$$

The log-likelihood of a multivariate TPPs is:

$$\log p(S) = \sum_{n=1}^{N} \log \lambda_{k_n}^*(t_n) - \int_0^T \lambda^*(t)dt, \tag{1}$$

where $\lambda^*(t) = \sum_{k=1}^{K} \lambda_k^*(t)$ is the ground intensity. We can estimate the model parameters by maximizing the log-likelihood in Equation (1). Additionally, given the history $\mathcal{H}_{t_n}$, the next event timestamp $t_{n+1}$'s probability density function can be written as:

$$p(t_{n+1} \mid \mathcal{H}_{t_n}) = \lambda^*(t_{n+1}) \exp\left(-\int_{t_n}^{t_{n+1}} \lambda^*(t)dt\right).$$

We can predict the next event timestamp $\widehat{t}_{n+1}$ and mark $\widehat{k}_{n+1}$ using the following estimator:

$$\widehat{t}_{n+1} = \int_{t_n}^{\infty} tp(t \mid \mathcal{H}_{t_n})dt, \quad \widehat{k}_{n+1} = \arg\max_k \frac{\lambda_k^*(\widehat{t}_{n+1})}{\lambda^*(\widehat{t}_{n+1})}. \tag{2}$$

### 2.2 CCIF PARAMETERIZATION

While CIF-based parameterization is commonly used for modeling TPPs, a major drawback is that the CIF integral in Equation (1) generally lacks a closed-form solution, requiring inefficient numerical integration (Omi et al., 2019). An alternative method is to directly model the CCIF $\Lambda_k^*(t) = \int_0^t \lambda_k^*(\tau)d\tau$, allowing us to rewrite Equation (1) as:

$$\log p(S) = \sum_{n=1}^{N} \log \frac{d}{dt} \Lambda_{k_n}^*(t_n^-) - \Lambda^*(T), \tag{3}$$

where $\Lambda^*(t) = \sum_{k=1}^{K} \Lambda_k^*(t)$, and $\frac{d}{dt}\Lambda_{k_n}^*(t_n^-)$ represents the left-hand derivative of $\Lambda_k^*(t)$ at $t_n, k_n$. By modeling the CCIF, MLE no longer requires integral computations but only differentiation, which can be easily and precisely handled using automatic differentiation (Baydin et al., 2018).

## 2.3 MONOTONE SPLINES

Monotone splines are a widely used method for modeling monotonic data and have been extensively applied to cumulative distribution function modeling (Durkan et al., 2019). Fritsch & Carlson (1980) introduced a technique for constructing $C^1$ monotone functions using piecewise cubic polynomials. Gregory & Delbourgo (1982) later developed piecewise rational quadratic splines (RQS), which preserve $C^1$ monotonicity while remaining easily differentiable and analytically invertible. Other similar monotone splines include rational cubic splines (RCS) (Abbas et al., 2012) and rational linear splines (RLS) (Dolatabadi et al., 2020).

Piecewise monotone splines ensure smoothness at interpolation endpoints. Taking RQS as an example, given an interpolation interval $[w_{m-1}, w_m]$, the $m$-th piecewise RQS can be written as:

$$f_m(t) = y_{m-1} + \frac{(y_m - y_{m-1})\left(s_m \tau^2 + \delta_{m-1}\tau(1-\tau)\right)}{s_m + (\delta_m + \delta_{m-1} - 2s_m)\tau(1-\tau)}, \tag{4}$$

where $\delta_{m-1}, \delta_m > 0$, $y_m > y_{m-1} > 0$, $s_m = \frac{y_m - y_{m-1}}{w_m - w_{m-1}}$ and $\tau = \frac{t - w_{m-1}}{w_m - w_{m-1}} \in [0, 1]$. It can be verified that $y_{m-1}, y_m, \delta_{m-1}, \delta_m$ are $f_m$'s function values and derivatives at interpolation endpoints. Specifically,

$$f_m(w_{m-1}) = y_{m-1}, \quad f_m(w_m) = y_m, \quad f_m'(w_{m-1}) = \delta_{m-1}, \quad f_m'(w_m) = \delta_m. \tag{5}$$

Equation (4) implies that given an interpolation interval $[w_{m-1}, w_m]$, a piecewise RQS can be uniquely defined by the endpoint values $y_{m-1}, y_m$ and derivatives $\delta_{m-1}, \delta_m$. If $f_{m-1}$ and $f_m$ share the same $y_{m-1}$ and $\delta_{m-1}$, two piecewise RQS can be seamlessly connected, ensuring the smoothness of the spline. RCS and RLS possess similar properties, which are referenced in Section C.

## 3 METHODOLOGY

In this section, we first analyze the requirements for modeling CCIF and discuss the limitations of existing MNN-based approaches. We then introduce a novel CCIF modeling method: MAS, and detail its implementation. For clarity, we begin by focusing on the univariate TPP before extending the discussion to multivariate TPPs.

## 3.1 MODELING CCIF AFTER CERTAIN EVENTS

A CCIF $\Lambda^*(t)$ can be decomposed into:

$$\Lambda^*(t) = \sum_{n=0}^{N} I_{(t_n, t_{n+1}]}\Lambda^{*(n)}(t), \tag{6}$$

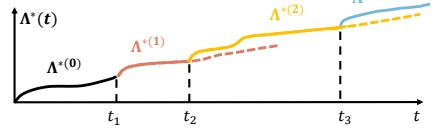

Figure 1: Modeling $\Lambda^*(t)$.

where $t_0 = 0$, $t_{N+1} = T$, and $I_{(t_n, t_{n+1}]}$ is an indicator function that takes the value 1 if $t \in (t_n, t_{n+1}]$ and 0 otherwise, $\Lambda^{*(n)}(t) = \int_{t_n}^{t} \lambda^*(\tau)\,d\tau + \Lambda^{*(n-1)}(t_n)$, whose support is $(t_n, \infty)$, representing the CCIF following the $n$-th event with $\Lambda^{*(-1)}(0) = 0$. It is worth noting that the support of $\Lambda^{*(n)}(t)$ is $(t_n, \infty)$ because, during prediction, the next event $t_{n+1}$ is unknown and can happen at any time in the range $(t_n, \infty)$. However, during training, since $t_{n+1}$ is known, we apply the indicator function $I_{(t_n, t_{n+1}]}$ to truncate $\Lambda^{*(n)}(t)$ to $(t_n, t_{n+1}]$. We then concatenate $\Lambda^{*(n)}(t)$ and $\Lambda^{*(n+1)}(t)$ to construct the complete $\Lambda^*(t)$, as is shown in Figure 1. The objective of this work is to propose a novel, flexible, and efficient approach for modeling $\Lambda^{*(n)}(t)$.

## 3.2 REQUIREMENTS FOR MODELING CCIF

As discussed in the introduction, modeling the CCIF of a TPP simplifies MLE and improves efficiency; however, CCIF is challenging to model due to its various requirements. Generally speaking, the following conditions are necessary for modeling common CCIF $\Lambda^*(t)$:

**(1) Global Monotonicity:** For any $0 < t_1 \leq t_2 < \infty$, it holds that $\Lambda^*(t_1) \leq \Lambda^*(t_2)$, $\Lambda^*(0) = 0$.

**(2) $C^1$-Continuity Between Timestamps:** For any $t_n < t < t_{n+1}$, $\Lambda^*(t)$ is continuous with a well-defined and continuous derivative $\Lambda^{*\prime}(t)$.

**(3) $C^0$-Continuity at Timestamps:** At each timestamp $t_n$, $\Lambda^*(t_n^-) = \Lambda^*(t_n^+)$, but the derivatives $\Lambda^{*\prime}(t_n^-)$ and $\Lambda^{*\prime}(t_n^+)$ may not be equal.

The second and third conditions ensure that the CIF remains continuous between any two consecutive timestamps but may exhibit jumps at the timestamps, which is a common characteristic of history-dependent TPP, such as the Hawkes process (Hawkes, 1971) and the self-correcting process (Isham & Westcott, 1979). Furthermore, to ensure that $\Lambda^*(t)$ satisfies these conditions, it is sufficient for $\Lambda^{*(n)}(t)$ to satisfy monotonicity and $C^1$ continuity, because the third condition is automatically fulfilled when concatenating $\Lambda^{*(n)}(t)$ and $\Lambda^{*(n+1)}(t)$.

At present, most CCIF-based methods utilize MNNs to model CCIF, ensuring monotonicity (Liu, 2024; Omi et al., 2019). However, this approach has several limitations. First, while these methods guarantee monotonicity, they may overlook other CCIF requirements. For example, the FullyNN proposed by Omi et al. (2019) does not enforce $\Lambda^*(0) = 0$, and the use of the ReLU activation function violates the $C^1$ continuity between consecutive timestamps. Second, MNNs suffer from limited expressivity. While a simple MNN can be constructed by constraining all weights in an MLP to be positive (Sill, 1997), such MNNs are inherently convex, and their expressivity is severely constrained. (See Section A for details.) Finally, adapting more complex neural network architectures into MNNs can be extremely challenging, which reduces their architectural flexibility.

### 3.3 MONOTONE ALTERNATING SPLINES

For complex TPPs, the corresponding $\Lambda^{*(n)}(t)$ can be highly intricate. A natural idea is to divide $\Lambda^{*(n)}(t)$ into smaller segments that are easier to fit with simple elementary functions. Inspired by this idea, we propose Monotone Alternating Splines (MAS) to model $\Lambda^{*(n)}(t)$. MAS consists of two parts: the interpolation part and the extrapolation part:

$$\Lambda_\theta^{*(n)}(t) = \underbrace{\sum_{m=1}^{M} I_{(w_{m-1}^{(n)}, w_m^{(n)}]} f_{m,\theta}^{*(n)}(t)}_{\text{MAS interpolation}} + \underbrace{I_{(w_M^{(n)}, \infty)} g_\theta^{*(n)}(t)}_{\text{MAS extrapolation}}, \tag{7}$$

where the support of $\Lambda^{*(n)}(t)$, i.e., $(t_n, \infty)$, is partitioned into $M + 1$ segments $(w_0^{(n)}, w_1^{(n)})$, ..., $(w_{M-1}^{(n)}, w_M^{(n)})$, $(w_M^{(n)}, \infty)$ with $w_0^{(n)} = t_n$, $f_{m,\theta}^{*(n)}(t)$ is the interpolated function over $(w_{m-1}^{(n)}, w_m^{(n)}]$, $g_\theta^{*(n)}(t)$ is the extrapolated function over $(w_M^{(n)}, \infty)$, and $\theta$ denotes the model parameters. We next discuss the implementation of each part in MAS.

### 3.3.1 MAS INTERPOLATION

To enforce $\Lambda_\theta^{*(n)}(t)$ satisfies monotonicity and $C^1$ continuity, the functions $f_{m,\theta}^{*(n)}(t)$ and $g_\theta^{*(n)}(t)$ must be monotonically increasing and $C^1$ continuous within each segment, which implies:

$$f_{m,\theta}^{*(n)}(w_m^{(n)}) = f_{m+1,\theta}^{*(n)}(w_m^{(n)}), \quad f_{m,\theta}^{*(n)\prime}(w_m^{(n)}) = f_{m+1,\theta}^{*(n)\prime}(w_m^{(n)}),$$
$$f_{M,\theta}^{*(n)}(w_M^{(n)}) = g_\theta^{*(n)}(w_M^{(n)}), \quad f_{M,\theta}^{*(n)\prime}(w_M^{(n)}) = g_\theta^{*(n)\prime}(w_M^{(n)}). \tag{8}$$

As is illustrated in Equation (5), monotone splines perfectly satisfy both monotonicity within each interpolation interval and $C^1$ continuity at the partition points. Following Equation (4), the monotone spline is determined when interpolation knots $\{w_m^{(n)}\}_{m=1}^M$, function values $\{\Lambda^{*(n)}(w_m^{(n)})\}_{m=1}^M$, and derivatives $\{\Lambda^{*(n)\prime}(w_m^{(n)})\}_{m=1}^M$ are given. In MAS, these parameters derive from historical information. We first employ an encoder (e.g., a Transformer, parameterized by $\theta_1$) to compress the history into an embedding $\mathbf{h}_n$. Then, an MLP (parameterized by $\theta_2$) followed by a softplus function maps $\mathbf{h}_n$ to the required parameters, yielding $f_{m,\theta}^{*(n)}(t)$, where $\theta = \{\theta_1, \theta_2\}$. We let $f_{m,\theta}^{*(n)}$ and $f_{m-1,\theta}^{*(n)}$ share the same endpoint derivative $\delta_{m-1}^{(n)}$ to ensure the $C^1$ continuity of $\Lambda_\theta^{*(n)}(t)$.

Furthermore, we set $\Lambda_\theta^{*(n)}(w_0^{(n)}) = \Lambda_\theta^{*(n)}(t_n) = \Lambda_\theta^{*(n-1)}(t_n)$ to ensure the seamless concatenation of $\Lambda_\theta^{*(n)}(t)$ and $\Lambda_\theta^{*(n-1)}(t)$ at $t_n$, as is illustrated in Figure 2.

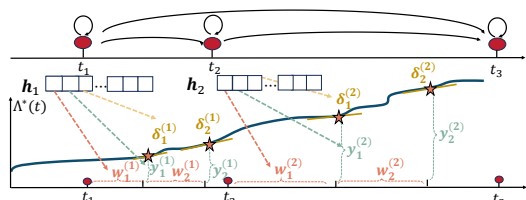

**Fitting Ability of MAS Interpolation**: Shorter interpolation intervals allow the MAS interpolation to approximate the ground truth $\Lambda^{*(n)}(t)$ more accurately. Formally, we obtain the following theorem.

Figure 2: MAS interpolation implementation.

**Theorem 3.1.** *Suppose the MAS interpolation support can extend to infinity, i.e., $w_M^{(n)} \to \infty$. Then, for any ground-truth $\Lambda^*(t)$, there exists an MAS representation $\Lambda_\theta^*(t) = \sum_{n=0}^N \mathcal{I}_{(t_n, t_{n+1}]} \Lambda_\theta^{*(n)}(t)$, such that:*

$$\|\Lambda^*(t) - \Lambda_\theta^*(t)\|_0 \le \frac{1}{2} C_0 \Delta^2, \tag{9}$$

*where $\|f(t)\|_0 = \sup_{t>0} |f(t)|$, $\Delta$ is the maximum interpolation interval, and $C_0$ is a constant independent of the data distribution. Furthermore, there exists an $\Lambda_\theta^*(t)$, such that,*

$$\|\Lambda^{*\prime}(t) - \Lambda_\theta^{*\prime}(t)\|_0 \le C_0 \Delta, \tag{10}$$

*where, at the timestamp $t_n$, the above derivatives are taken as the left-hand derivatives.*

Theorem 3.1 states that MAS fits any ground-truth CIF and CCIF provided that the interpolation intervals are sufficiently small, which implies its fitting performance is comparable to any other TPP parameterization.

### 3.3.2 MAS EXTRAPOLATION

Theorem 3.1 assumes that the MAS interpolation support can extend to infinity, i.e., $w_M^{(n)} \to \infty$. However, in practice, this is not possible. Therefore, an additional monotonic extrapolation function $g_\theta^{*(n)}(t)$ must be used to model $\Lambda^{*(n)}(t)$ over $(w_M^{(n)}, \infty)$. It is worth noting that most monotone splines (without any additional tail design) are based on polynomials, which do not guarantee mono-

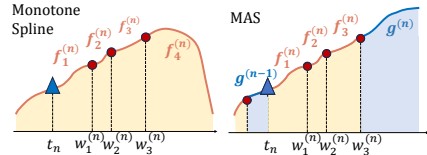

Figure 3: MAS and the monotone spline.

tonicity outside the interpolation interval. Taking the RQS in Equation (4) as an example, when $s_m^{(n)}$ is smaller than $\delta_{m-1}^{(n)}$ and $t$ is sufficiently large, $f_m^{(n)}(t)$ becomes monotonic decreasing and can even take negative values. See Figure 3 for an illustration. Thus, an alternative parameterization is required for $g_\theta^{*(n)}(t)$ in Equation (7) to ensure the global monotonicity of $\Lambda^{*(n)}(t)$ over $(w_M^{(n)}, \infty)$.

In implementation, $g_\theta^{*(n)}(t)$ can be a linear function $g_\theta^{*(n)}(t) = at + b$ with $a, b > 0$, or an exponential function $g_\theta^{*(n)}(t) = b - \exp(-at)$ with $a, b > 0$. Like MAS interpolation, the extrapolation's parameters $a, b$ are also obtained from the history embedding $\mathbf{h}_n$. Again, $\mathbf{h}_n$ derives from an encoder (e.g., a Transformer, parametrized by $\theta_1$) and is converted to $a, b$ using an MLP (parametrized by $\theta_2$) followed by a softplus function. The extrapolation $g_\theta^{*(n)}(t)$ is then characterized by $\theta = \{\theta_1, \theta_2\}$.

Notably, when concatenating MAS's interpolation and extrapolation, smoothness at the junction point $(w_M^{(n)})$ is always guaranteed. To ensure smoothness, we first determine the extrapolation function $g_\theta^{*(n)}(t)$ through history embedding. Then, we set $f_{M,\theta}^{*(n)}(w_M^{(n)}) = g_\theta^{*(n)}(w_M^{(n)})$ and $f_{M,\theta}^{*(n)\prime}(w_M^{(n)}) = g_\theta^{*(n)\prime}(w_M^{(n)})$. According to Equation (4), $f_{M,\theta}^{*(n)}$ can be uniquely determined. Consequently, the $C^1$ continuity of $\Lambda^{*(n)}(t)$ at $w_M^{(n)}$ can always be satisfied.

**Equivalence to Classical TPPs**: As an important hyperparameter in Equation (7), a larger $w_M^{(n)}$ increases the contribution of piecewise monotone spline interpolation, enhancing the model's fitting performance. Conversely, a smaller $w_M^{(n)}$ amplifies the influence of the extrapolation component,

improving the model's generalization performance (further discussed in Section 4). Notably, when $w_M^{(n)} = t_n$, MAS reduces to classical TPP models.

**Corollary 3.2.** *A homogeneous Poisson process with intensity $\mu$ can be expressed by a MAS with linear extrapolation, where $w_M^{(n)} = t_n$ and $g^{*(n)}(t) = \mu t, t \in [t_n, \infty)$.*

**Corollary 3.3.** *An exponential-kernel Hawkes process with the intensity function $\lambda^*(t) = \mu + \sum_{t_n < t} \alpha \exp(-\beta(t - t_n))$ can be expressed by a MAS with the sum of linear and exponential extrapolations, where $w_M^{(n)} = t_n$ and $g^{*(n)}(t) = \mu t + \frac{\alpha}{\beta} \sum_{i=1}^{n} (1 - \exp(-\beta(t - t_i))), t \in [t_n, \infty)$.*

Similar to the two corollaries above, other classical TPPs can also be represented within the MAS framework. By incorporating the piecewise monotone spline interpolation component, MAS extends classical TPPs and significantly enhances the model's flexibility.

**Efficiency of MAS**: During the model training process using MLE, MAS does not require automatic differentiation to compute derivatives at each timestamp in Equation (3), since all monotone splines have analytical derivatives. While automatic differentiation is necessary for MNN-based methods, MAS gets rid of the automatic differentiation step and further improves computational efficiency. In Section C.1, we provide a derivation of the analytical derivative for RQS.

### 3.4 MAS FOR MULTIVARIATE TPPS

MAS can be extended to multivariate TPPs. If the multivariate TPPs consist of $K$ different event types, we can use $K$ MAS models to represent $K$ CCIFs. Given the event history $\{(t_1, k_1), \ldots, (t_n, k_n)\}$, a Transformer (parameterized by $\theta_1$) is employed to encode the historical information into a history embedding $\mathbf{h}_n$. Then, $\mathbf{h}_n$ is mapped by $K$ different MLPs followed by softplus (parameterized by $\theta_{2,1}, \ldots, \theta_{2,K}$) to $K$ sets of distinct variables for $K$ MAS models. Consequently, we obtain $\Lambda_1^*(t), \ldots, \Lambda_K^*(t)$. The overall model parameters are given by $\theta = \{\theta_1, \theta_{2,1}, \ldots, \theta_{2,K}\}$. Subsequently, we train the model using Equation (3) and utilize Equation (2) to predict the next timestamp and event type. It is worth noting that $K$ MAS models share the same history $\mathbf{h}_n$, which implies that the $K$ CCIFs influence each other. This interplay is a crucial property in multivariate TPPs.

## 4 GENERALIZATION ANALYSIS

This section explores the generalization of MAS. Suppose the training set consists of $Z$ i.i.d. point process sequences $S_1, \ldots, S_Z$ on the time window $[0, T]$. The $n$-th timestamp of the $z$-th sequence is denoted as $t_{zn}$. For simplicity, we consider the univariate case, where the max sequence length is $N_0$. Without loss of generality, we assume the interpolation interval is a constant $\Delta > 0$. The empirical loss is defined as the averaged log-likelihood over all sequences: $\widehat{\mathcal{L}} := \frac{1}{Z} \sum_{z=1}^{Z} \frac{1}{T} \log p(S_z)$. Though we train the model by minimizing $\widehat{\mathcal{L}}$, we are interested in the expected model performance over the data distribution. In other words, we want to bound the training loss's expectation: $\mathcal{L} := \mathbb{E}[\widehat{\mathcal{L}}]$. We can first obtain the following lemma.

**Lemma 4.1.** *Given a sequence $S_z = \{t_{z1}, \cdots, t_{zN}\}$, $N \leq N_0$ and a MAS model $\Lambda_\theta^*(t)$ determined by parameter $\theta$ or $\eta$, the loss $\log p_\theta(S_z)$ is Lipschitz continuous w.r.t. $\theta$:*

$$\|\log p_\theta(S_z) - \log p_\eta(S_z)\| \leq \frac{C_1}{T} \sum_{n=1}^{N} r_n \|\theta - \eta\|,$$

*where $r_n = m$ if $w_{m-1}^{(n)} \leq t_{zn} \leq w_m^{(n)}$, and $C_1$ is a constant independent of data distribution.*

Then, we can derive the error bound between $\mathcal{L}$ and $\widehat{\mathcal{L}}$.

**Theorem 4.2.** *Suppose there are $M$ knots within a MAS. Then, with probability $1 - \xi$, $|\widehat{\mathcal{L}} - \mathbb{E}[\widehat{\mathcal{L}}]| \leq \frac{1}{\sqrt{Z}} \left( \frac{1}{2} \sqrt{\log \frac{1}{\xi}} + C_2 \sqrt{N_0 M} \int_0^c \sqrt{C_3 - \log t} dt \right)$, where $c, C_2, C_3$ are data-independent constants.*

Theorem 4.2 offers the bound between the empirical and generalized loss, implying minimizing the empirical loss $\widehat{\mathcal{L}}$ helps reduce the generalized loss $\mathbb{E}[\widehat{\mathcal{L}}]$. Taking a step forward, we combine Theorem 3.1 and Theorem 4.2 to derive the upper bound for MAS's generalized error.

**Theorem 4.3.** *Suppose the total interpolation length of MAS is a fixed constant $L$, and each interval of the monotone spline has length $\Delta$. Then, with probability at least $1 - \xi$, the following bound holds:*

$$\mathbb{E}[\widehat{\mathcal{L}}] \leq \mathcal{L}^* + \underbrace{\frac{1}{2\sqrt{Z}}\sqrt{\log\frac{1}{\xi}}}_{\text{probability error}} + \underbrace{\frac{B}{T \cdot L}}_{\text{extrapolation error}} + \underbrace{C_0\left(\frac{\Delta}{C_4} + \frac{\Delta^2}{2}\right)}_{\text{interpolation error}} + \underbrace{\frac{C_2}{\sqrt{Z}}\frac{\sqrt{L}}{\sqrt{\Delta}}R(L, \Delta)}_{\text{complexity error}}, \quad (11)$$

*where $\mathcal{L}^*$ is the global optimal loss for the ground-truth CCIF, $B, C_0, C_2, C_4$ are constants independent of sampling randomness, $R(L, \Delta)$ is a constant increasing as $L$ increase and $\Delta$ decreases.*

*Remark* 4.4. Without loss of generality, the interpolation interval $\Delta$ is set to a constant in our generalization analysis. When implementing MAS, each interpolation knot $w_m^{(n)}$ can be variable and determined by the history information embedding $\mathbf{h}_n$. Even so, Theorem 4.3 still holds — one only needs to replace $\Delta$ with the upper bound on the interpolation-interval length.

In Theorem 4.3, besides the probability error, the remaining terms can be divided into three parts, each caused by the interpolation, extrapolation, and model complexity. First, the **interpolation error** $C_0\left(\frac{\Delta}{C_4} + \frac{\Delta^2}{2}\right)$ decreases as the interpolation interval $\Delta$ shrinks, as a smaller $\Delta$ leads to a more accurate fit to the ground-truth CCIF. Second, the **extrapolation error** $\frac{B}{T \cdot L}$ decreases as the total interpolation length $L$ increases. As the contribution of the interpolation part becomes larger, the influence of the extrapolation part diminishes. Finally, the **complexity error** $\frac{C_2}{\sqrt{Z}}\frac{\sqrt{L}}{\sqrt{\Delta}}R(L, \Delta)$ increases as $\Delta$ decreases and $L$ increases, and can be reduced by increasing the sample size $Z$. If we expand $L$ and reduce $\Delta$ (i.e., stronger interpolation), the number of parameters in MAS increases, leading to higher model complexity. Consequently, more samples are needed to ensure the model's generalization performance.

Theorem 4.3 offers guidance on selecting hyperparameters. Given a fixed sample size $Z$, two important hyperparameters, $\Delta$ and $L$, can be estimated by minimizing the bound in Equation (11), as is presented in the following corollary.

**Corollary 4.5.** *In Theorem 4.3, when the generalization bound is minimized, the total interpolation length $L$ and interpolation interval $\Delta$ satisfy:*

$$L \geq \sqrt{\frac{\mathbb{E}[t_i] \cdot \lambda_{min}}{l_1}}, \quad \Delta \leq \frac{(Z)^{\frac{1}{3}}(\mathbb{E}[t_i])^{\frac{5}{6}}\sqrt{\lambda_{min}}}{2\sqrt{l_1}q^{\frac{1}{3}}(3N_0)^{\frac{1}{3}}},$$

*where $\mathbb{E}[t_i]$ represents the expectation of the event interval, $q$ is the freedom of each interpolation knot (commonly $q = 3$), $\lambda_{min}$ is the base intensity during $[0, T]$, and $l_1$ is the Lipschitz constant of the intensity during event intervals. Specifically, for any $\tau_1, \tau_2$ s.t. $t_{z,i} < \tau_1 < \tau_2 < t_{t,i+1}$, $|\lambda(\tau_1) - \lambda(\tau_2)| \leq l_1|\tau_1 - \tau_2|$.*

**Selection of $L$ and $\Delta$:** According to Theorem 4.5, hyperparameters $L$ and $\Delta$ depend on two dataset-specific quantities: (1) the expected inter-event time $\mathbb{E}[t_i]$, and (2) the degree of temporal variability in the TPP intensity, captured by its Lipschitz constant $l_1$. As $\mathbb{E}[t_i]$ grows, a longer interpolation horizon, i.e., a larger $L$, is required to track changes in the CCIF accurately. Moreover, as the intensity fluctuation, reflected by $l_1$, becomes sharper, MAS should provide a finer-grained approximation, which requires a smaller interpolation $\Delta$. Furthermore, Theorem 4.5 provides a simple plug-in strategy for selecting $L$ and $\Delta$: For synthetic datasets where the functional form of the intensity is known, $l_1, \lambda_{min}$, and $\mathbb{E}[t_i]$ can be estimated directly from samples, yielding an approximate lower bound for $L$ and $\Delta$. For general real-world datasets, inspired by Scott's rule (Hollander et al., 2013), we suggest assuming that the data follow a classic TPP, e.g., exponential-kernel Hawkes process, and estimating $L$ and $p$ accordingly. Given the estimated $L$ and $\Delta$, the number of interpolation knots $p$ can also be calculated as $p = L/\Delta$.

It is worth noting that, although Theorem 4.3 shows that an excessively large $L$ and an overly small $\Delta$ may lead to overfitting and thus increased generalization error, such overfitting can be effectively mitigated by standard training techniques (e.g., dropout (Srivastava et al., 2014) or SGD-based optimization (Keskar & Socher, 2017)). In practice, we observe that this overfitting effect is mild. Therefore, we recommend using a moderately large $L$ and a relatively fine $\Delta$, consistent with the bounds provided in Theorem 4.5.

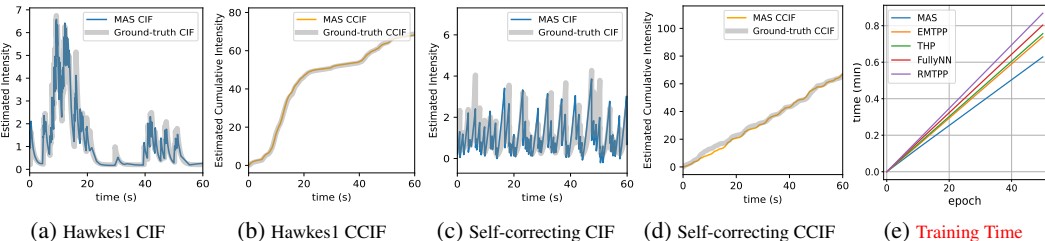

Figure 4: Performance of MAS. (a)–(d): Estimated MAS intensity and cumulative intensity on the Hawkes1 and Self-correcting datasets. (e): The training time of MAS, FullyNN and EMTPP.

(a) Hawkes1 CIF    (b) Hawkes1 CCIF    (c) Self-correcting CIF    (d) Self-correcting CCIF    (e) Training Time

Table 1: Performance comparison between baselines and MAS on the univariate synthetic datasets with respect to NLL and RMSE. Both metrics are better when smaller. The best metric is highlighted in bold, and the second-best metric is underlined. Standard deviations are reported in parentheses.

| MODEL | HAWKES1 | | HAWKES2 | | RENEWAL1 | | RENEWAL2 | | SELF-CORRECTING | |
|---|---|---|---|---|---|---|---|---|---|---|
| | NLL($\downarrow$) | RMSE($\downarrow$) | NLL($\downarrow$) | RMSE($\downarrow$) | NLL($\downarrow$) | RMSE($\downarrow$) | NLL($\downarrow$) | RMSE($\downarrow$) | NLL($\downarrow$) | RMSE($\downarrow$) |
| IPP | -1.010 $_{(0.001)}$ | 2.391 $_{(0.019)}$ | -0.992 $_{(0.006)}$ | 2.613 $_{(0.005)}$ | -1.491 $_{(0.002)}$ | 1.976 $_{(0.025)}$ | -0.809 $_{(0.001)}$ | 3.028 $_{(0.015)}$ | -1.281 $_{(0.002)}$ | 1.144 $_{(0.019)}$ |
| HAWKES | -1.645 $_{(0.009)}$ | 2.365 $_{(0.002)}$ | -1.553 $_{(0.0092)}$ | 2.523 $_{(0.004)}$ | -1.336 $_{(0.007)}$ | 2.143 $_{(0.013)}$ | 1.487 $_{(0.012)}$ | 3.382 $_{(0.009)}$ | -1.123 $_{(0.008)}$ | 1.282 $_{(0.011)}$ |
| RMTPP | -1.472 $_{(0.023)}$ | 2.387 $_{(0.006)}$ | -1.880 $_{(0.021)}$ | 2.583 $_{(0.004)}$ | -1.571 $_{(0.039)}$ | 2.071 $_{(0.012)}$ | -1.950 $_{(0.030)}$ | 3.064 $_{(0.007)}$ | -1.348 $_{(0.021)}$ | 1.162 $_{(0.001)}$ |
| THP | -1.512 $_{(0.010)}$ | 2.453 $_{(0.018)}$ | -1.534 $_{(0.028)}$ | 2.637 $_{(0.052)}$ | -1.619 $_{(0.001)}$ | 1.710 $_{(0.002)}$ | -1.508 $_{(0.016)}$ | 2.980 $_{(0.085)}$ | -1.312 $_{(0.001)}$ | 1.221 $_{(0.005)}$ |
| SAHP | -1.766 $_{(0.029)}$ | 2.460 $_{(0.017)}$ | -2.102 $_{(0.001)}$ | 2.645 $_{(0.053)}$ | -1.888 $_{(0.007)}$ | 1.737 $_{(0.001)}$ | -1.973 $_{(0.009)}$ | 2.981 $_{(0.089)}$ | -1.421 $_{(0.001)}$ | 1.122 $_{(0.019)}$ |
| FULLYNN | -1.471 $_{(0.014)}$ | 2.463 $_{(0.033)}$ | -1.488 $_{(0.021)}$ | 2.651 $_{(0.018)}$ | -1.648 $_{(0.011)}$ | 1.929 $_{(0.057)}$ | -1.453 $_{(0.011)}$ | 3.064 $_{(0.039)}$ | -1.374 $_{(0.006)}$ | 1.220 $_{(0.018)}$ |
| TRITPP | -1.510 $_{(0.005)}$ | 2.409 $_{(0.022)}$ | -1.939 $_{(0.000)}$ | 2.628 $_{(0.007)}$ | -1.821 $_{(0.001)}$ | 2.038 $_{(0.027)}$ | -1.910 $_{(0.002)}$ | 3.081 $_{(0.028)}$ | -1.328 $_{(0.002)}$ | 0.897 $_{(0.017)}$ |
| EMTPP | -1.466 $_{(0.021)}$ | 2.324 $_{(0.049)}$ | -1.468 $_{(0.013)}$ | 2.541 $_{(0.007)}$ | -1.646 $_{(0.004)}$ | 1.671 $_{(0.102)}$ | -1.453 $_{(0.012)}$ | 3.038 $_{(0.014)}$ | -1.378 $_{(0.003)}$ | 1.201 $_{(0.061)}$ |
| IFTPP | -1.802 $_{(0.026)}$ | 2.429 $_{(0.032)}$ | -1.873 $_{(0.007)}$ | 2.563 $_{(0.052)}$ | -1.903 $_{(0.009)}$ | 1.789 $_{(0.012)}$ | -1.385 $_{(0.004)}$ | 2.967 $_{(0.104)}$ | -1.478 $_{(0.002)}$ | 0.963 $_{(0.009)}$ |
| WSM | -1.552 $_{(0.0114)}$ | 2.438 $_{(0.015)}$ | -1.578 $_{(0.016)}$ | 2.609 $_{(0.041)}$ | -1.599 $_{(0.001)}$ | 1.681 $_{(0.119)}$ | -1.531 $_{(0.016)}$ | 2.943 $_{(0.116)}$ | -1.308 $_{(0.001)}$ | 1.219 $_{(0.000)}$ |
| DTPP | -1.455 $_{(0.005)}$ | 2.951 $_{(0.007)}$ | -2.031 $_{(0.012)}$ | 2.517 $_{(0.002)}$ | -1.651 $_{(0.007)}$ | 1.833 $_{(0.018)}$ | -2.097 $_{(0.011)}$ | 2.875 $_{(0.046)}$ | -1.341 $_{(0.003)}$ | 0.898 $_{(0.000)}$ |
| CUFUN | -1.488 $_{(0.006)}$ | 2.441 $_{(0.009)}$ | -1.543 $_{(0.011)}$ | 2.634 $_{(0.005)}$ | -1.844 $_{(0.003)}$ | 1.441 $_{(0.002)}$ | -1.454 $_{(0.005)}$ | 3.069 $_{(0.023)}$ | -1.410 $_{(0.002)}$ | 1.220 $_{(0.001)}$ |
| HYPRO | -1.239 $_{(0.003)}$ | 2.361 $_{(0.011)}$ | -0.948 $_{(0.005)}$ | 2.553 $_{(0.021)}$ | -1.524 $_{(0.004)}$ | 1.712 $_{(0.082)}$ | -1.521 $_{(0.012)}$ | 3.012 $_{(0.012)}$ | -1.214 $_{(0.004)}$ | 1.192 $_{(0.002)}$ |
| MAS | **-1.807** $_{(0.024)}$ | **2.302** $_{(0.025)}$ | **-2.203** $_{(0.010)}$ | **2.448** $_{(0.059)}$ | **-1.955** $_{(0.003)}$ | 1.479 $_{(0.178)}$ | **-2.097** $_{(0.012)}$ | **2.869** $_{(0.041)}$ | -1.371 $_{(0.006)}$ | 1.109 $_{(0.054)}$ |

## 5 EXPERIMENTS

This section presents the performance of MAS in both univariate and multivariate scenarios, compared to other baseline models. We also evaluate the efficiency of MAS relative to other CCIF- or CIF-based methods. An ablation study is conducted at the end of the experiment.

**Baselines**: We compare our method against thirteen representative baselines. First, we include a simple inhomogeneous Poisson process (**IPP**) (Daley & Vere-Jones, 2003) and a univariate parametric Hawkes process. Next, we consider three models based on the conditional intensity function (CIF): the RNN-based **RMTPP** (Du et al., 2016), the attention-based **THP** (Zuo et al., 2020), and the self-attentive **SAHP** (Zhang et al., 2020). We also include four models based on the cumulative intensity function (CIF-integrated form): **FullyNN** (Omi et al., 2019), **TriTPP** (Shchur et al., 2020b), **EMTPP** (Liu, 2024), and **CuFun** (Wang et al., 2024b). Finally, we include **IFTPP** (Shchur et al., 2020a), an intensity-free method, and **WSM** (Cao et al., 2025), an integral-free method leveraging score matching, and other recent SOTA TPP models, including **HYPRO** (Xue et al., 2022) and **DTPP** (Panos, 2024).

Some baselines, such as IPP, Hawkes, FullyNN, and TriTPP, only consider the univariate case in their original implementations. Therefore, our experiments are divided into two parts. First, we compare the performance of MAS with all baselines on univariate synthetic data. Then, we evaluate MAS on multivariate real-world datasets, comparing it with baselines that support multivariate settings, including RMTPP, THP, SAHP, EMTPP, IFTPP, WSM, CuFun, HYPRO, and DTPP.

**Metrics**: We use three evaluation metrics: negative log-likelihood (**NLL**), root-mean-square error (**RMSE**), and accuracy (**ACC**). For NLL, we compute the log-likelihood on the test data according to Equation (1) and Equation (3). For RMSE, we predict the next timestamp using Equation (2) and calculate the RMSE between the predicted timestamp $\widehat{t}_{n+1}$ and the ground-truth timestamp $t_{n+1}$. For ACC, we predict the next event type based on Equation (2) and compute the average accuracy.

**Experimental Setup**: For history-dependent baselines, some models use RNNs to extract history embeddings, such as RMTPP, while others use Transformers, such as THP and SAHP. To ensure a fair

Table 2: Performance comparison between baselines and MAS on the multivariate real-world datasets with respect to NLL, ACC and RMSE. The best metric is highlighted in bold, the second-best underlined. Standard deviations are reported in parentheses.

| MODEL | RETWEET | | | EARTHQUAKE | | | TAXI | | | TAOBAO | | |
|---|---|---|---|---|---|---|---|---|---|---|---|---|
| | NLL($\downarrow$) | ACC($\uparrow$) | RMSE($\downarrow$) | NLL($\downarrow$) | ACC($\uparrow$) | RMSE($\downarrow$) | NLL($\downarrow$) | ACC($\uparrow$) | RMSE($\downarrow$) | NLL($\downarrow$) | ACC($\uparrow$) | RMSE($\downarrow$) |
| RMTPP | -0.468 (0.011) | 0.559 (0.001) | 22.19 (0.003) | 0.251 (0.006) | 0.424 (0.000) | 1.530 (0.004) | -0.299 (0.059) | 0.905 (0.005) | 0.366 (0.001) | -0.191 (0.029) | 0.442 (0.001) | 0.132 (0.000) |
| THP | -0.459 (0.003) | 0.585 (0.000) | 20.88 (0.004) | 0.300 (0.010) | 0.451 (0.000) | 1.391 (0.005) | -0.293 (0.001) | 0.913 (0.001) | 0.356 (0.000) | -0.227 (0.025) | 0.594 (0.021) | 0.132 (0.000) |
| SAHP | -0.454 (0.000) | 0.584 (0.000) | **20.43** (0.019) | 0.275 (0.012) | 0.417 (0.003) | 1.448 (0.001) | -0.442 (0.077) | 0.902 (0.005) | 0.358 (0.002) | -0.586 (0.099) | 0.572 (0.011) | 0.141 (0.003) |
| EMTPP | -0.446 (0.012) | 0.595 (0.001) | 22.10 (0.075) | 0.388 (0.009) | 0.458 (0.002) | 1.517 (0.006) | -0.312 (0.025) | 0.915 (0.003) | 0.362 (0.001) | -0.039 (0.005) | 0.591 (0.008) | 0.131 (0.001) |
| IFTPP | -0.477 (0.015) | **0.603** (0.003) | 22.18 (0.204) | 0.191 (0.034) | 0.434 (0.013) | 1.488 (0.007) | -0.206 (0.019) | 0.914 (0.006) | 0.377 (0.003) | -0.594 (0.013) | 0.446 (0.005) | 0.134 (0.003) |
| WSM | -0.449 (0.002) | 0.599 (0.000) | 22.12 (0.017) | 0.235 (0.002) | **0.470** (0.000) | 1.510 (0.003) | -0.281 (0.001) | 0.916 (0.001) | 0.361 (0.001) | **-1.115** (0.053) | 0.581 (0.005) | 0.132 (0.000) |
| DTPP | -0.459 (0.002) | 0.601 (0.001) | 22.38 (0.012) | 0.256 (0.015) | 0.462 (0.002) | 1.405 (0.041) | -0.446 (0.012) | 0.923 (0.001) | 0.361 (0.007) | -0.984 (0.003) | 0.593 (0.005) | 0.128 (0.000) |
| CuFun | -0.462 (0.001) | 0.600 (0.002) | 20.96 (0.008) | 0.230 (0.012) | 0.463 (0.001) | 1.430 (0.014) | -0.273 (0.001) | 0.914 (0.002) | 0.345 (0.004) | -0.878 (0.012) | 0.585 (0.005) | 0.133 (0.003) |
| HYPRO | -0.411 (0.023) | 0.599 (0.002) | 21.24 (0.003) | 0.743 (0.032) | 0.456 (0.001) | 1.452 (0.012) | -0.428 (0.011) | 0.914 (0.005) | 0.363 (0.003) | -0.947 (0.032) | 0.583 (0.004) | 0.130 (0.002) |
| MAS | **-0.489** (0.044) | 0.597 (0.005) | 20.75 (0.673) | **0.186** (0.007) | 0.468 (0.004) | **1.281** (0.013) | **-0.474** (0.003) | **0.925** (0.002) | **0.332** (0.001) | -1.003 (0.055) | **0.600** (0.021) | **0.125** (0.003) |

comparison, we employ the same Transformer as the history encoder for both the Transformer-based baselines and our proposed MAS. For MAS, we incorporate 10 knots for the interpolation component, using RQS as the interpolation monotone spline and an exponential function $g(t) = a - \exp(-bt)$ as the extrapolation function (other spline and extrapolation functions are analyzed in the ablation study). The interpolation support $L$ is set to a constant. For all baselines, hyperparameters are set according to the default settings from their original repositories. All event sequences are randomly split into 60% training data, 20% validation data, and 20% test data. All models are trained with the Adam optimizer, conducted on an RTX 4090 with 24GB of memory. Details refer to Section D.

## 5.1 SYNTHETIC UNIVARIATE DATA

**Datasets**: In the synthetic univariate experiment, we consider five datasets constructed by Omi et al. (2019): two Hawkes processes (1,2) with different intensities, two renewal processes 1,2 (stationary/non-stationary), and a self-correcting process. Details refer to Section D.

**Results**: Table 1 presents the performance comparison between all baselines and MAS. MAS achieved the best performance across most metrics on the majority of datasets. This validates our theory that, by introducing piecewise monotone spline as the interpolation component, MAS is capable of fitting various ground-truth CCIFs derived from different classical TPP. In contrast, other traditional models (e.g., IPP) or deep Hawkes processes (e.g., THP) become less effective in characterizing other TPP. Meanwhile, MAS also outperforms models that parametrize CCIF using MNNs in terms of fitting ability. Figure 4 shows MAS's predicted CIF and CCIF for the two datasets, implying MAS successfully captures the intensity patterns of different TPP. A detailed cross-validation between MAS and FullyNN is conducted in Section D.3, further illuminating the superiority of MAS over other classic CCIF-based methods.

## 5.2 REAL MULTIVARIATE DATA

**Datasets**: We consider four real multivariate datasets: **Taxi** Whong (2014), **Taobao** Xue et al. (2023), **Retweet** Zhou et al. (2013) and **Earthquake** Xue et al. (2023). Detailed information is in Section D.

**Results**: Table 2 presents the performance comparison between multivariate baselines and MAS. On real-world datasets, MAS once again achieved the best results across multiple metrics. This further demonstrates the advantages of MAS in terms of flexibility and generalization, indicating that in multivariate scenarios, MAS successfully captures the interactions among multiple variables.

The success of MAS can be attributed to two factors: first, the inclusion of more interpolation knots along the timeline, which makes MAS more flexible than baseline models; second, the introduction of the extrapolation component, which ensures that MAS correctly simulates the CCIF while maintaining strong generalization ability. Therefore, our model achieves greater flexibility compared to baseline models without compromising generalization.

## 5.3 EFFICIENCY, LONG HORIZON PREDICTION, AND ABLATION STUDY

**Efficiency**: We compare the efficiency of MAS with THP, RMTPP, and two CCIF-based models: EMTPP and FullyNN. Using the Hawkes-1 dataset, we ensure that the number of model parameters

is nearly the same across all three models and measure the wall-clock time required to run 50 epochs under identical experimental conditions. As is presented in Figure 4e, MAS is significantly faster than the four baseline models, as it eliminates automatic differentiation. The details are provided in Section D.3.2.

**Long Horizon Prediction**: We conduct long-horizon prediction experiments on a univariate dataset (Hawkes1) and a multivariate dataset (TAXI). We compare MAS against THP and two CCIF multivariate modeling methods (CuFun and EMTPP). For each model, we require it to predict the next 3/5/10 events. The results are shown in Table 3. As can be seen, MAS not only achieves high accuracy in single-step prediction but also maintains its superiority in long-horizon prediction.

Table 3: Long-horizon results.

| | TAXI | | HAWKES1 |
|---|---|---|---|
| HORIZON | ACC%(↑) | RMSE(↓) | RMSE(↓) |
| | **THP** | | |
| NEXT 3 | 0.818 | 0.585 | 3.597 |
| NEXT 5 | **0.796** | 0.832 | 5.445 |
| NEXT 10 | 0.780 | 1.338 | 8.356 |
| | **EMTPP** | | |
| NEXT 3 | 0.818 | 0.588 | 3.921 |
| NEXT 5 | 0.794 | 0.740 | 5.129 |
| NEXT 10 | 0.750 | 1.173 | 7.995 |
| | **CUFUN** | | |
| NEXT 3 | 0.802 | **0.523** | 3.833 |
| NEXT 5 | 0.790 | 0.758 | 5.146 |
| NEXT 10 | 0.772 | 1.341 | 7.797 |
| | **MAS** | | |
| NEXT 3 | **0.819** | 0.525 | **3.312** |
| NEXT 5 | 0.791 | **0.704** | **4.281** |
| NEXT 10 | **0.785** | **1.005** | **6.235** |

Table 4: The results of ablation study.

| | TAXI | | EARTHQUAKE | |
|---|---|---|---|---|
| | NLL(↓) | ACC%(↑) | NLL(↓) | ACC%(↑) |
| **NUMBER OF INTERPOLATION KNOTS** | | | | |
| P=5 | -0.460 | 91.90 | 0.185 | **46.70** |
| P=10 | **-0.474** | **92.50** | 0.183 | 46.57 |
| P=50 | -0.467 | 92.10 | **0.180** | 46.42 |
| P=100 | -0.447 | 91.90 | 0.209 | 44.07 |
| **INTERPOLATION LENGTH** | | | | |
| L=4 | -0.427 | 91.70 | 0.190 | 45.70 |
| L=6 | **-0.474** | **92.50** | 0.183 | 46.57 |
| L=10 | -0.424 | 92.40 | **0.172** | **46.60** |
| **EXTRAPOLATION FUNCTION** | | | | |
| EXP | **-0.474** | **92.50** | 0.183 | **46.57** |
| LINEAR | -0.460 | 92.00 | **0.180** | 46.50 |
| **INTERPOLATION FUNCTION** | | | | |
| RQS | **-0.474** | 92.50 | **0.183** | 46.57 |
| RLS | -0.470 | **93.00** | 0.186 | **46.60** |

**Ablation Study**: We conduct ablation studies to verify the contribution of each component in MAS. We consider the following factors: **(1)** the type of interpolation monotone spline (RQS/RLS), **(2)** the type of extrapolation function (linear/exponential), **(3)** the interpolation support length ($L = 4/6/10$), **(4)** the number of interpolation knots ($p = 5/10/50/100$). The experiments are performed on the Taxi and Earthquake datasets, using NLL and ACC as metrics. We set RQS interpolation, exponential extrapolation, $L = 6$, and $p = 10$ as the default choices.

As shown in Table 4, as the number of interpolation knots $p$ increases and the interpolation support length $L$ extends, the model's performance on both datasets first improves and then declines. This validates our theoretical analysis: the increase in interpolation support length and the number of interpolation knots leads to a rise in the complexity error term in Theorem 4.3. This overfitting, however, is not severe. Furthermore, MAS is not sensitive to different interpolation or extrapolation functions. We speculate that, under default settings (i.e., $L = 6$ and $p = 10$), MAS can already fit the CCIFs of both datasets at a fine granularity, so different functions yield no significant difference. More ablation study results are provided in Section D.3.3.

## 6    CONCLUSIONS

We propose a novel approach, MAS, for modeling the CCIF of TPPs. Compared to MNNs, MAS offers greater generality, flexibility, and efficiency, and enjoys strong theoretical guarantees for both fitting and generalization. We prove that the interpolation component of MAS enhances fitting performance, while the extrapolation component of MAS enhances generalization performance. Extensive experiments demonstrate that MAS outperforms other CCIF- and CIF-based modeling methods on both univariate and multivariate datasets across multiple evaluation metrics.

ETHICS STATEMENT

Our work adheres to the ICLR Code of Ethics. Our proposed method, MAS, is intended for research purposes to better capture inner dynamics in stochastic events, and we foresee no direct negative societal consequences from its use.

REPRODUCIBILITY STATEMENT

We have made every effort to ensure the reproducibility of our research. The complete source code for our proposed method is included in the supplementary materials. Additionally, detailed descriptions of the experimental setup, baselines, and evaluation metrics are provided in the experiment section to facilitate replication of our results.

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

## A  RELATED WORK

Before MAS, a few TPP models were developed to model the cumulative conditional intensity function (CCIF). However, these methods often relied on monotonic neural networks (MNNs) for modeling. For example, Omi et al. (2019) employed a simple MNN—by constraining all weights in a multilayer perceptron (MLP) to be positive—to model the cumulative intensity function. Let $\boldsymbol{h}^{(l)}$ denote the hidden state at layer $l$, the forward propagation of an $L$-layer positive-weighted MLP can be defined as:

$$\Lambda(t) = \boldsymbol{W}^{(L)}\sigma\left(\cdots\sigma\left(\boldsymbol{W}^{(2)}\sigma\left(\boldsymbol{W}^{(1)}t + \boldsymbol{b}^{(1)}\right) + \boldsymbol{b}^{(2)}\right)\right) + \boldsymbol{b}^{(L)}. \tag{12}$$

where $\sigma(\cdot)$ is the activation function, and $\boldsymbol{W}^{(l)}$ is the model weights. Its derivative regarding $t$ is:

$$\lambda(t) = \frac{\partial\Lambda(t)}{\partial t} = \boldsymbol{W}^{(L)}\left(\prod_{l=L-1}^{1} \text{diag}\left[\sigma'(\boldsymbol{z}^{(l)})\right]\boldsymbol{W}^{(l)}\right). \tag{13}$$

Since common activation functions (e.g., ReLU, Softplus, Tanh) are non-decreasing, their derivatives satisfy $\sigma'(\cdot) \geq 0$. Thus, by enforcing the MLP parameter $\boldsymbol{W}^{(l)} \geq 0$, the intensity function $\lambda(t)$ modeled by this MNN remains positive. Liu (2024) extended FullyNN to multivariate TPPs, but still used the same positive weighted-MLP architecture. A similar approach is used in modeling the conditional cumulative distribution function (Wang et al. (2024a)), which also relies on the same MNN architecture to ensure monotonicity. A main drawback of the positive-weighted MLP is its theoretically limited expressivity. Typically, MNNs use activation functions such as ReLU or softplus; however, these functions are usually convex. Their linear combinations in an MNN are therefore also convex with respect to $t$. As a result, the CCIF modeled by such MNN-based approaches is *inherently convex*, making it unable to capture more flexible CCIF shapes. For example, in an exponentially decaying Hawkes process, the intensity function is monotonically decreasing—in other words, the second derivative of the CCIF is negative, so the CCIF is non-convex. In this case, an MNN cannot properly model the final decay behavior of the Hawkes process.

Apart from the CCIF and the conditional cumulative distribution function, score-matching methods are also introduced to avoid numerical integration in TPP. However, many methods only work for specific point processes. Cao et al. (2025) proposes weighted score matching (WSM), which can be applied to more general point processes.

Monotonic splines are at the core of MAS. Splines like RQS and RLS have also been widely used in normalizing flows (Durkan et al., 2019). Shchur et al. (2020b) proposed viewing TPP sequences as a differentiable transformation of a homogeneous Poisson process, and used splines such as RQS to model this transformation. However, TriTPP may not be well-suited for processes such as Hawkes, due to computational efficiency issues and the choice of triangular maps. On the one hand, a triangular map cannot easily capture the recursive Markov structure in Hawkes process intensities, and can only approximate Hawkes using a triangular map with $O(NH)$ complexity. On the other hand, when the triangular map uses RQS monotonic splines, the transformation defaults to a linear interpolation $y = x$ outside the transformation window, assuming that the intensity remains constant over long horizons, rather than decaying exponentially back to the baseline as in Hawkes processes. We believe these issues make TriTPP struggle to fit Hawkes processes well in practice (as shown in both the original paper and our experiments). In contrast, our MAS model chooses a Transformer as a history encoder and adopts different tail functions (in addition to the linear tail, we also include exponential tails, etc.), enabling MAS to effectively model various types of TPPs, including Hawkes processes.

## B  NOTATIONS

All notations used in the paper are listed in the Table 5.

## C  MONOTONE SPLINES

This appendix section introduces some common monotonic splines and discusses their parameterizations.

Table 5: List of notations.

| Symbol | Definition | Symbol | Definition |
|---|---|---|---|
| $a_m, b_m \cdots e_m$ | parameters of $f_m(t)$ | $C_1, \cdots C_3, B$ | constants |
| $\delta_m, y_m, w_m$ | gradient, height, width of the $m$-th interpolation point | $\Delta$ | maximum interpolation interval |
| $f_m(t)$ | $m$-th interpolation in monotone splines | $f_{m,\theta}^{*(n)}$ | $m$-th MAS's interpolation after $t_n$, controlled by $\theta$ |
| $g(t)$ | an extrapolation function | $g_\theta^{*(n)}$ | MAS's extrapolation after $t_n$, controlled by $\theta$ |
| $\boldsymbol{h}_n$ | history embedding | $k_n$ | $n$-th event type |
| $L$ | interpolation length | $\lambda_k^*$ | ground-truth conditional intensity function |
| $\lambda^{*(n)}$ | ground-truth conditional intensity after event $n$ | $\Lambda_k^*$ | conditional cumulative intensity for the $k$-th variable |
| $\Lambda^{*(n)}$ | ground-truth conditional cumulative intensity after $t_n$ | $\Lambda_g^*$ | estimated cumulative intensity, controlled by $\theta$ |
| $\mathcal{L}$ | generalization error | $\widehat{\mathcal{L}}$ | empirical error |
| $\mathcal{L}^*$ | global optimal error from the ground-truth CCIF | $M$ | total interpolation points |
| $N$ | the number of timestamps in a sequence | $N_0$ | maximum sequence length |
| $p(\cdot)$ | probability measure | $S$ | a TPP sequence |
| $T$ | Time window | $t_n$ | $n$-th timestamp |
| $\theta_1$ | parameters in the encoder | $\theta_2$ | parameters in the MLP |
| $\widehat{t}_{n+1}$ | predicted event timestamp | $\widehat{k}_{n+1}$ | predicted event type |
| $\mathcal{H}_{t^-}$ | past history | $Z$ | number of sequences |

## C.1 RATIONAL QUADRATIC SPLINES

An piece of RQS on the interval $[w_{m-1}, w_m]$ can be written as:

$$f_m(t) = y_{m-1} + \frac{(y_m - y_{m-1})\left(s_m \tau^2 + \delta_{m-1}\tau(1-\tau)\right)}{s_m + (\delta_m + \delta_{m-1} - 2s_m)\,\tau(1-\tau)},$$

where $s_m = \frac{y_m - y_{m-1}}{w_m - w_{m-1}}$ and $\tau = \frac{t - w_{m-1}}{w_m - w_{m-1}} \in (0, 1]$. When $t = w_{m-1}$, i.e., $\tau = 0$, $f_m(t) = y_{m-1}$, $f'_m(t) = \delta_{m-1}$. When $t = w_m$, i.e., $\tau = 1$, $f_m(t) = y_m$, $f'_m(t) = \delta_m$. This implies that when $w_{m-1}, w_m, \delta_{m-1}, \delta_m, y_{m-1}, y_m$ are decided, the RQS can be uniquely decided.

Its derivative can be written as:

$$f'_m(\tau) = (y_m - y_{m-1})\frac{\left[2s_m \tau + \delta_{m-1}(1 - 2\tau)\right]}{J_m}$$
$$- (y_m - y_{m-1})\frac{\left[s_m \tau^2 + \delta_{m-1}\,\tau(1-\tau)\right]V_m(1 - 2\tau)}{J_m^2},$$

where $V_m = \delta_m + \delta_{m-1} - 2s_m$, $J_m = \left[s_m + V_m\,\tau(1-\tau)\right]$.

## C.2 RATIONAL CUBIC SPLINES

An piece of RCS on the interval $[w_{m-1}, w_m]$ can be written as:

$$f_m(t) = \frac{(1-\tau)^3 v_m y_m + \tau(1-\tau)^2\left[(2u_m v_m + v_m)y_m + v_m h_m d_m\right]}{(1-\tau)^2 v_m + 2u_m v_m \tau(1-\tau) + \tau^2 u_m}$$
$$+ \frac{\tau^2(1-\tau)\left[(2u_m v_m + u_m)y_{m+1} - u_m h_m \delta_{m+1}\right] + \tau^3 u_m y_{m+1}}{(1-\tau)^2 v_m + 2u_m v_m \tau(1-\tau) + \tau^2 u_m}.$$

where $\tau = \frac{t - w_{m-1}}{w_m - w_{m-1}} \in (0, 1]$ and $h_i = w_m - w_{m-1}$, $f_m(w_{m-1}) = y_{m-1}$, $f_m(w_m) = y_m$, $f'_m(w_{m-1}) = \delta_{m-1}$, $f'_m(w_m) = \delta_m$. $u_m$ and $v_m$ are two shape parameter, $u_m, v_m > 0$.

## C.3 MONOTONE CUBIC SPLINES

On $[w_i, w_{i+1}]$, let $h_i = w_{i+1} - w_i$, $\tau = \frac{t - w_i}{h_i}$, then a piece of monotone cubic spline can be expressed as:

$$f_m(t) = y_m\, h_{00}(\tau) + h_m\,\delta_m\, h_{10}(\tau)$$
$$+ y_{m+1}\, h_{01}(\tau) + h_m\,\delta_{m+1}\, h_{11}(\tau),$$

where

$$h_{00}(\tau) = 2\tau^3 - 3\tau^2 + 1, \quad h_{10}(\tau) = \tau^3 - 2\tau^2 + \tau,$$
$$h_{01}(\tau) = -2\tau^3 + 3\tau^2, \quad h_{11}(\tau) = \tau^3 - \tau^2.$$

Again, $f_m(w_{m-1}) = y_{m-1}$, $f_m(w_m) = y_m$, $f'_m(w_{m-1}) = \delta_{m-1}$, $f'_m(w_m) = \delta_m$.

## C.4 RATIONAL LINEAR SPLINES

On the interval $[w_{m-1}, w_m]$, a piece of RLS can be written as:

$$f(t) = a_m + \frac{c_m(t - w_m)}{1 + d_i(t - w_m)}, \quad x \in [w_{m-1}, w_m]. \tag{14}$$

$f(t)$ is monotone-increasing as long as $c_m > 0$.

## D EXPERIMENTS

### D.1 DATASETS

We consider the following five univariate datasets:

**(1) Hawkes Process (1, 2)**: We consider two exponential-decay Hawkes processes with CIF: $\lambda^*(t) = 0.2 + \sum_{t_n < t} \sum_{i=1}^M \alpha_i \exp(-\beta_i(t - t_n))$. We set $M = 1, \alpha_1 = 0.8, \beta_1 = 1$ for the first dataset and $M = 2, \alpha_1 = 0.4, \beta_1 = \beta_2 = 1, \alpha_2 = 0.4$ and $\beta_2 = 20$ for the second dataset.

**(2) Self-correcting Process**: The CIF is given as $\lambda^*(t) = \exp(t - \sum_{t_n < t} 1)$.

**(3) Renewal Process (Stationary / Non-stationary)**: For the stationary case, timestamp intervals are independent, log-normal distributed with a mean $1.0$ and standard deviation $0.5$. For the non-stationary case, the trend function is set to $r(t) = 0.99 \sin(2\pi t/20000) + 1$.

For multivariate datasets, their information is listed as follows:

**(1) Taxi** Whong (2014): This dataset tracks the time-stamped taxi pick-up and drop-off events across the New York City. The event types are categorized into $K = 10$ types and there are 2000 sequences with an average sequence length of 40.

**(2) Taobao** Xue et al. (2023): This dataset contains time-stamped user click behaviors on Taobao shopping pages. The event types are categorized into $K = 20$ types and there are 4800 sequences with an average sequence length of 150.

**(3) Retweet** Zhou et al. (2013): This dataset contains time-stamped user retweet event sequences. The events are categorized into $K = 3$ types. There are 9000 sequences with an average sequence length of 90.

**(4) Earthquake** Xue et al. (2023): This dataset contains time-stamped earthquake event sequences. The events are categorized into $K = 7$ types. There are 3000 sequences with an average sequence length of 20.

### D.2 EXPERIMENTAL SETUP

Because MNN is sensitive to the scale of its outputs, we applied a timestamp transformation to part of our data. Following the approach of Cao et al. (2025), we divided every timestamp in the retweet dataset by 100 and divided every timestamp in the earthquake dataset by 5. In addition, we divided the timestamps of each of the five synthetic datasets by 10. This ensures that all models converge within 100 epochs.

All models were trained on a single NVIDIA RTX·4090·D (24 GB). For fairness, MAS, FullyNN, EMTPP, THP, and WSM all encode historical information using Transformers of the same size. The Transformer hyperparameters are as follows: number of attention heads = 16, number of layers = 1, model dimension = 64, inner feed-forward dimension = 8, key dimension = 16, value dimension = 16, and dropout rate = 0.1. For MAS, when using a Rational Quadratic Spline (RQS) as the interpolation function, we set the lower bound for both the interpolation interval width and length to 0.01, and the minimum derivative at interpolation points to 0.01.

We fixed the batch size at 64 and the learning rate at 0.001 for all models. Each model was trained for 100 epochs on the train set and converged within 100 epochs. We then evaluated their performance on the test set.

Table 6: Cross-validation results.

| Softplus Activation (Size 800) | | | Sigmoid Activation (Size 800) | | |
|---|---|---|---|---|---|
| Configuration | NLL | RMSE | Configuration | NLL | RMSE |
| *FullyNN (positive MLP layers)* | | | *FullyNN (positive MLP layers)* | | |
| 1 layer MLP size 16 | -0.774 | 2.347 | 1 layer MLP size 16 | -0.774 | 2.347 |
| 1 layer MLP size 32 | -0.782 | 2.378 | 1 layer MLP size 32 | -0.782 | 2.378 |
| 2 layer MLP size 16 | -1.452 | 2.462 | 2 layer MLP size 16 | -1.437 | 2.453 |
| 2 layer MLP size 32 | -1.399 | 2.463 | 2 layer MLP size 32 | -1.449 | 2.449 |
| 3 layer MLP size 16 | -1.371 | 2.461 | 3 layer MLP size 16 | -1.443 | 2.469 |
| 3 layer MLP size 32 | -1.282 | 2.459 | 3 layer MLP size 32 | -1.418 | 2.458 |
| MAS | -1.807 | 2.302 | MAS | -1.807 | 2.302 |

| Softplus Activation (Size 400) | | | Sigmoid Activation (Size 400) | | |
|---|---|---|---|---|---|
| Configuration | NLL | RMSE | Configuration | NLL | RMSE |
| *FullyNN (positive MLP layers)* | | | *FullyNN (positive MLP layers)* | | |
| 1 layer MLP size 16 | -0.778 | 2.317 | 1 layer MLP size 16 | -0.778 | 2.317 |
| 1 layer MLP size 32 | -0.779 | 2.318 | 1 layer MLP size 32 | -0.779 | 2.318 |
| 2 layer MLP size 16 | -1.431 | 2.464 | 2 layer MLP size 16 | -1.347 | 2.448 |
| 2 layer MLP size 32 | -1.389 | 2.446 | 2 layer MLP size 32 | -1.404 | 2.490 |
| 3 layer MLP size 16 | -1.327 | 2.458 | 3 layer MLP size 16 | -1.311 | 2.497 |
| 3 layer MLP size 32 | -1.200 | 2.475 | 3 layer MLP size 32 | -1.325 | 2.425 |
| MAS | -1.779 | 2.297 | MAS | -1.779 | 2.297 |

When comparing the accuracy of MAS with that of other models, we observed that—even for the same model on the same dataset—the reported performance can vary dramatically across different studies. For common baseline models such as SAHP, THP, and RMTPP, we referenced their best-reported accuracy from the literature (for example, Zhang et al. (2024), Xue et al. (2023), and Cao et al. (2025)) and our experiments, and compared those to our MAS. For instance, on the Taobao dataset, THP achieved an accuracy of 0.467 according to Zhang et al. (2024), 0.531 according to Xue et al. (2023), and 0.594 in Cao et al. (2025). all below MAS's performance of 0.600.

For datasets like Taobao, where the raw time intervals are very large, we first scaled them down by a factor of 1/100 to ensure rapid convergence of models such as EMTPP. However, this scaling can cause the model's NLL to change substantially, making it difficult to compare against past work (since they may have used different scaling factors). Therefore, we retrained all baseline models on the same scaled dataset—keeping the data, training and testing strategy, and number of epochs identical—and then reported their NLL on the same test set.

### D.3    SUPPLEMENTARY RESULTS

#### D.3.1    CROSS VALIDATION

To rigorously demonstrate that MAS indeed outperforms other baselines, we conduct a cross-validation study comparing MAS with a classical CCIF-based method, FullyNN. We evaluate FullyNN under different network depths, widths, activation functions, and training data sizes to ensure that MAS's superiority is not merely due to accidental hyperparameter choices.

Specifically, we fix the encoder architecture of MAS and FullyNN to be identical (i.e., both models receive the same input features). Following the data generation procedure of the hawkes1 dataset, we independently sample 400 and 800 sequences within the time range $[0, 100]$ for training. During training, we vary FullyNN's number of fully connected layers from 1 to 3, its hidden widths between 16 and 32, and its activation functions between softplus and sigmoid (representing different levels of smoothness). FullyNN is trained for 100 epochs until convergence. We then compare its NLL and RMSE with those of MAS, as shown in Table 6.

The results show that when the network depth exceeds two layers, the performance of FullyNN no longer improves significantly with additional depth. Moreover, regardless of whether softplus or sigmoid is used, MAS consistently outperforms FullyNN across all configurations. This cross-validation experiment further confirms the advantage of MAS over classical CCIF-based modeling methods.

### D.3.2 TRAINING TIME

We align the parameter sizes of five models—THP, RMTPP, MAS, EMTPP, and FullyNN—and train them for 100 epochs on the Hawkes1 dataset. Experiments are rerun three times under the same conditions. We then measure the mean and the standard deviation of their running time (in minutes), and the results are summarized in the table below. A visualized version of this comparison is shown in Figure 4e.

Table 7: Comparison of running time.

|  | MAS | EMTPP | THP | FullyNN | RMTPP |
|---|---|---|---|---|---|
| 10 epochs | 0.128 (0.001) | 0.145 (0.000) | 0.151 (0.001) | 0.163 (0.001) | 0.173 (0.001) |
| 50 epochs | 0.629 (0.001) | 0.739 (0.001) | 0.756 (0.001) | 0.803 (0.001) | 0.868 (0.002) |

### D.3.3 EXTRA ABLATION STUDY

We also conduct additional ablation experiments to examine the role of the tail function further. We fix the number of interpolation knots at $p = 10$ and use RQS as the interpolation function, while separately adopting a linear tail or an exponential tail as the extrapolation function. We additionally control the total interpolation length to be $L = 4$ or $L = 10$, and train the models on the Hawkes1 dataset as well as on a time-homogeneous Poisson process. The training procedure follows the same settings as in the main experiments.

The results in Table 8 show that, on the Poisson process, MAS with a linear tail performs slightly better than with an exponential tail, whereas the exponential tail yields better performance on the Hawkes process. Moreover, under the Poisson & exponential-tail configuration, increasing the total interpolation length improves model performance. This is consistent with the following intuition: when the inter-event time exceeds the interpolation length, the intensity function becomes fully determined by the extrapolation. In this regime, if the dataset exhibits Hawkes-like dynamics, i.e., self-excitation followed by gradual decay, an exponential tail is more suitable. In contrast, if the dataset behaves more like a Poisson process, where the intensity tends to stabilize toward a constant value, a linear tail is a better choice.

Moreover, although different extrapolation functions in MAS adapt to different data characteristics, the choice typically leads to only mild differences, especially when the interpolation part is sufficiently expressive to capture most of the TPP intensity variation, as shown in Table 8.

Table 8: Comparison of linear and exponential tail functions.

| Tail Function | $L$ Length | Hawkes | | Poisson | |
|---|---|---|---|---|---|
| | | NLL | RMSE | NLL | RMSE |
| Linear | $L = 4$ | -1.713 | 2.323 | -1.293 | 1.006 |
| | $L = 6$ | -1.733 | 2.317 | -1.302 | 1.024 |
| Exponential | $L = 4$ | -1.758 | 2.297 | -1.272 | 1.051 |
| | $L = 6$ | -1.807 | 2.302 | -1.286 | 1.048 |

## E PROOF

The following mild assumptions are required to prove the theorems in the paper.

**Assumption E.1** ($C^1$-Lipschitz Continuity). The ground-truth CCIF and MAS's interpolation possess Lipschitz continuous derivative w.r.t time $t$ and parameter $\theta$ on any fixed closed interpolation interval between timestamps. The Lipschitz constants are $l_1$ and $l_2$ respectively.

**Assumption E.2** (Positive Derivative). The ground-truth CCIF and MAS' interpolation possesses positive derivative: $\Lambda_\theta^{*\prime}, \Lambda^{*\prime} > 0$.

### E.1 Proof of Theorem 3.1

*Proof.* We start from $\Lambda_\theta^{*\prime}(t)$. When the interpolation length $w_M^{(n)}$ is sufficiently large, only the interpolation function affects MAS. For common monotone splines, given the endpoint values and derivatives, the spline parameter can be uniquely defined. Then, for any interval $[a, b] \subset [t_n, t_{n+1}]$ for some timestamp $t_n$, set

$$\Lambda_\theta^{*\prime}(a) = \Lambda^{*\prime}(a), \Lambda_\theta^*(a) = \Lambda^*(a),$$
$$\Lambda_\theta^{*\prime}(b) = \Lambda^{*\prime}(b), \Lambda_\theta^*(b) = \Lambda^*(b).$$

Given Theorem E.1, $f'$ is continuous on $[a, b]$. Meanwhile, based on Theorem E.2, $f'$ is upper and lower bounded: $\exists v_1, v_2 > 0, v_1 < f' < v_2$. Thus, for any $t \in [a, b]$, the following inequality holds:

$$|\Lambda_\theta^{*\prime}(t) - \Lambda^{*\prime}(t)| \le |\Lambda_\theta^{*\prime}(t) - \Lambda_\theta^{*\prime}(a)| + |\Lambda^{*\prime}(t) - \Lambda^{*\prime}(a)| \le 2l_1|t - a|.$$

Note that the maximum interpolation length is $\Delta$. The above inequality implies $\|\Lambda_\theta^{*\prime} - \Lambda^{*\prime}\| \le 2l_1\Delta$, which is the first inequality in Theorem 3.1, where $C_0 = 2l_1$.

Meanwhile,

$$|\Lambda_\theta^*(t) - \Lambda^*(t)| = |\int_a^t \Lambda^{*\prime}(w) - \Lambda_\theta^{*\prime}(w)dw|$$
$$\le \int_a^t |\Lambda^{*\prime}(w) - \Lambda_\theta^{*\prime}(w)|dw$$
$$\le \int_a^t C_0|t - a|dw \le \frac{1}{2}C_0t^2 \le \frac{1}{2}C_0\Delta^2.$$

Thus, we derive the fitting error bound for $\Lambda_\theta^*(t)$ and $\Lambda_\theta^{*\prime}(t)$. $\qquad\square$

### E.2 Proof of Theorem 4.2

When proving the generalization bound of the MAS model, we further assume that the horizontal interval of MAS interpolation is a fixed value $\Delta$. That is, after each timestamp $t_n$, there are $N$ interpolated points, with a total length of $L = N\Delta$. For the next timestamp $t_{n+1}$, there is a probability that it falls into one of the $M$ interpolation intervals, or into the extrapolation interval $[t_n + N\Delta, \infty)$. We denote the index of the interval into which $t_{n+1}$ falls as $r_{n+1}$, taking values in $1, 2, \ldots, M$.

We first prove Theorem 4.1:

*Proof.* For each time series, the loss function can be decomposed into the loss over the $N, N \le N_0$ timestamps.

$$\log p(S) = \sum_{n=0}^{N-1} \log \frac{d}{dt}\Lambda_{k_n}^*(t_{n+1}) - \Lambda^*(T)$$
$$= \sum_{n=0}^{N-1} \log \frac{d}{dt}\Lambda_{k_n}^*(t_{n+1}) - (\Lambda^*(t_{n+1}) - \Lambda^*(t_n)) + \Lambda^*(T) - \Lambda^*(t_N)$$
$$:= \sum_{n=1}^{N} \log p(t_n) + \Lambda^*(T) - \Lambda^*(t_N),$$

where $t_{N+1} = T$. For simplicity, we omit $\Lambda^*(T) - \Lambda^*(t_N)$ in the above loss.

For different parameter $\theta$ and $\eta$, the difference in $\log p(t_n, \theta)$ and $\log p(t_n, \eta)$ can be written as:

$$\| \log p(t_n, \theta) - \log p(t_n, \eta) \|$$
$$= | \left( \log \left( \Lambda_\theta^{*\prime}(t_n) \right) - \left( \Lambda_\theta^*(t_n) - \Lambda_\theta^*(t_{n-1}) \right) \right) - \left( \log \left( \Lambda_\eta^{*\prime}(t_n) \right) - \left( \Lambda_\eta^*(t_n) - \Lambda_\eta^*(t_{n-1}) \right) \right) |$$
$$\leq \underbrace{\left| \log \left( \Lambda_\theta^{*\prime}(t_n) \right) - \log \left( \Lambda_\eta^{*\prime}(t_n) \right) \right|}_{T_1} + \underbrace{\left| \left( \Lambda_\theta^*(t_n) - \Lambda_\eta^*(t_{n-1}) \right) - \left( \Lambda_\theta^*(t_n) - \Lambda_\eta^*(t_{n-1}) \right) \right|}_{T_2}.$$

We next discuss the bound for $T_1$ and $T_2$. For $T_1$, we utilize the Lipschitz property in Theorem E.1:

$$T_1 = \left| \log \left( \Lambda_\theta^{*\prime}(t_{n+1}) \right) - \log \left( \Lambda_\eta^{*\prime}(t_{n+1}) \right) \right| \leq \frac{1}{v_1} \| \Lambda_\theta^{*\prime}(t_{n+1}) - \Lambda_\eta^{*\prime}(t_{n+1}) \|,$$

where $v_1$ is the lower bound for MAS's derivative w.r.t. the parameter $\theta$. Suppose $t_{n+1}$ falls in the $r_n$-th interval in MAS. Suppose the parameter $\theta = \{ \theta_1, \theta_2, \cdots \theta_M \}$. Each component $\theta_m$ is a sub-vector, deciding the height $\Lambda_\theta^*(t_i + (m-1)\Delta)$ and the derivative $\Lambda_\theta^{*\prime}(t_i + (m-1)\Delta)$ and further decides the interpolation $f_{m,\theta}^{*(n)}$. Because $t_n + \Delta \cdot (r_n - 1) < t_{n+1} < t_n + \Delta \cdot (r_n)$, according to Lipschitz-continuity in Theorem E.1, the above inequality can be written as:

$$T_1 \leq \frac{1}{v_1} \| \Lambda_\theta^{*\prime}(t_{n+1}) - \Lambda_\eta^{*\prime}(t_{n+1}) \| \leq \frac{l_2}{v_1} \| \theta^{r_n} - \eta^{r_n} \|.$$

We next bound $T_2$. Note that,

$$\Lambda_\theta^* (t_{n+1}) - \Lambda_\theta^* (t_n) = \left( \Lambda_\theta^* (t_n + \Delta) - \Lambda_\theta^* (t_n) \right) + \left( \Lambda_\theta^* (t_n + 2\Delta) - \Lambda_\theta^* (t_n + \Delta) \right)$$
$$+ \left( \Lambda_\theta^* (t_n + (r_n - 1)\Delta) - \Lambda_\theta^* (t_n + (r_n - 2)\Delta) \right)$$
$$+ \left( \Lambda_\theta^* (t_{n+1}) - \Lambda_\theta^* (t_n + (r_n - 1)\Delta) \right)$$
$$= \theta^1 + \theta^2 + \cdots + \theta^{r_n - 1} + \left( \Lambda_\theta^* (t_{n+1}) - \Lambda_\theta^* (t_n + (r_n - 1)\Delta) \right).$$

Thus we can acquire:

$$T_2 = \left| \left( \Lambda_\theta^*(t_n) - \Lambda_\eta^*(t_{n-1}) \right) - \left( \Lambda_\theta^*(t_n) - \Lambda_\eta^*(t_{n-1}) \right) \right|$$
$$= | \sum_{m=1}^{r_n} \theta^m + \left( \Lambda_\theta^* (t_{n+1}) - \Lambda_\theta^* (t_n + r_n \Delta) \right) - \sum_{m=1}^{r_n} \eta^m + \left( \Lambda_\eta^* (t_{n+1}) - \Lambda_\eta^* (t_n + r_n \Delta) \right) |$$
$$\leq \sum_{m=1}^{r_n} \| \theta^m - \eta^m \| + l_2 \| \theta^{r_n + 1} - \eta^{r_n + 1} \|.$$

Combining the bound for $T_1$ and $T_2$, we obtain that:

$$\| \log p(t_n, \theta) - \log p(t_n, \eta) \| \leq T_1 + T_2 \leq \frac{l_2}{v_1} \| \theta^{r_n} - \eta^{r_n} \| + \sum_{m=1}^{r_n - 1} | \theta^m - \eta^m | + l_2 \| \theta^{r_n} - \eta^{r_n} \|$$
$$\leq \left( \frac{l_2}{v_1} + l_2 \right) \| \theta^{r_n} - \eta^{r_n} \| + \sum_{m=1}^{r_n - 1} \| \theta^m - \eta^m \|$$
$$\leq \max \left( \frac{l_2}{v_1} + l_2, 1 \right) \sum_{m=1}^{r_n} \| \theta^m - \eta^m \|$$
$$\leq \max \left( \frac{l_2}{v_1} + l_2, 1 \right) r_n \| \theta - \eta \|.$$

By summing up $\log p(t_n)$ and taking avarage over time $T$, we can acquire:

$$\| \log p_\theta(S) - \log p_\eta(S) \| \leq \frac{C_1}{T} \sum_{n=1}^{N} r_n \| \theta - \eta \|,$$

where $C_1 = \max \left( \frac{l_2}{v_1} + l_2, 1 \right)$. This finishes the proof. $\qquad \square$

Before proving Theorem 4.2, we introduce the following concepts.

**Definition E.3** (bracketing number (Van der Vaart, 2000))**.** Given two functions $l$ and $u$, the *bracket* $[l, u]$ is the set of all functions $f$ with $l \leq f \leq u$. Given a probability measure $\mu$, an $\varepsilon$*-bracket* is a bracket $[l, u]$ with $\|u - l\|_{L_2} = \left( \int (u - l)^2 \mathrm{d}\mu \right)^{\frac{1}{2}} < \varepsilon$. The *bracketing number* $N_{[]}(\varepsilon, \mathcal{F})$ is the minimum number of $\varepsilon$-brackets needed to cover $\mathcal{F}$. Specifically,

$$N_{[]}(\epsilon, \mathcal{F}) = \arg\min_n \{[l_n, u_n], \|u_n - l_n\| \leq \epsilon, ds.t. \forall f \in \mathcal{F}, \exists j \in [n], l_j \leq f \leq u_j\}.$$

Bracketing numbers quantify how many simple function pairs (brackets) are needed to approximate a complex function class within a given accuracy. The smaller the bracketing number, the simpler or more regular the function class is. Utilizing the following lemma in (Van der Vaart, 2000), we can acquire the bracketing number bound for MAS:

**Lemma E.4.** *Let $\mathcal{F} = \{f_\theta : \theta \in \Theta\}$ be a collection of measurable functions indexed by a bounded subset $\Theta \subset \mathbb{R}^d$. Suppose that there exists a measurable function $m$ such that*

$$|f_{\theta_1}(x) - f_{\theta_2}(x)| \leq m(x)\|\theta_1 - \theta_2\|, \quad \text{for every } \theta_1, \theta_2.$$

*Then there exists a constant $K$, depending on $\Theta$ and $d$ only, such that the bracketing numbers satisfy*

$$N_{[\,]}(\varepsilon, \mathcal{F}) \leq K \left( \frac{\|m\|_{L_2} \operatorname{diam} \Theta}{\varepsilon} \right)^d \quad \text{for every } 0 < \varepsilon < \operatorname{diam} \Theta,$$

*where $\operatorname{diam} \Theta$ is the diameter of the parameter space $\Theta$.*

Utilizing Theorem E.4 and Theorem 4.1, the bracketing number of MAS can be upper bounded:

**Theorem E.5.** *The bracketing number of MAS is upper bounded. Suppose a set of MAS $\mathcal{M}$ is controlled by $d$-dimensional parameter $\theta \in \Theta \subset R^d$. For any $0 < \epsilon < \operatorname{diam} \Theta$,*

$$N_{[\,]}(\varepsilon, \mathcal{M}) \leq K \left( \frac{c_1 \operatorname{diam} \Theta}{\varepsilon} \right)^d, \tag{15}$$

*where $c_1 = \frac{B}{T} \left( \mathbb{E} \left( \sum_{n=1}^{N_0} r_n \right)^2 \right)^{\frac{1}{2}}$.*

An important feature of the bracketing number is that it bounds the complexity of a model. We first introduce the concept of Rademacher complexity (Koltchinskii, 2001).

**Definition E.6** (Rademacher complexity (Koltchinskii, 2001))**.** The *Rademacher complexity* of a class of functions $\mathcal{F}$ with respect to a sample $\{x_1, x_2, \ldots, x_z\}$ drawn from a distribution $P$ is defined as:

$$\mathcal{R}_z(\mathcal{F}) = \mathbb{E}_\sigma \left[ \sup_{f \in \mathcal{F}} \frac{1}{Z} \sum_{z=1}^{Z} \sigma_z f(x_z) \right],$$

where $\sigma_1, \sigma_2, \ldots, \sigma_z$ are independent Rademacher random variables, i.e., $\mathbb{P}(\sigma_z = 1) = \mathbb{P}(\sigma_z = -1) = \frac{1}{2}$, and the expectation $\mathbb{E}_\sigma$ is taken over the distribution of these random variables.

Rademacher complexity can be understood as the degree to which a model fits noise. The higher the Rademacher complexity of a model, the stronger its ability to fit noise, and the more prone it is to overfitting—i.e., the generalization error increases. We can derive the following lemma:

**Lemma E.7.** *For a real-valued function space $\mathcal{F} : \mathcal{X} \rightarrow [0, 1]$, given a training set $X = \{x_1, x_2, \ldots, x_Z\}$ of size $Z$ sampled independently and identically distributed from $\mathcal{D}$, for any $f \in \mathcal{F}$ and $0 < \xi < 1$, with probability at least $1 - \delta$, we have:*

$$\left[ \mathbb{E}[f(x)] - \frac{1}{Z} \sum_{z=1}^{Z} f(x_z) \right] \leq 2\mathcal{R}_Z(\mathcal{F}) + \sqrt{\frac{\ln(1/\xi)}{2Z}}.$$

There exists the following relationship between the Rademacher complexity and the bracketing number of the model:

**Theorem E.8** (Bounding Rademacher Complexity via Bracketing Number). *Let $\mathcal{F}$ be a class of measurable functions, and let $P$ be a probability distribution over $\mathcal{X}$. Suppose that the bracketing number $N_{[]}(\varepsilon, \mathcal{F})$ is finite for all $\varepsilon > 0$. Then, for a sample $S = \{x_1, \ldots, x_Z\}$ drawn i.i.d. from probability measure $\mu$, the empirical Rademacher complexity satisfies:*

$$\mathcal{R}_Z(\mathcal{F}) \leq \inf_{\delta > 0} \left( 4\delta + \frac{12}{\sqrt{Z}} \int_{\delta}^{\operatorname{diam} \Theta} \sqrt{\log N_{[]}(\varepsilon, \mathcal{F})} \, d\varepsilon \right)$$

$$\leq \frac{12}{\sqrt{Z}} \int_0^{\operatorname{diam} \Theta} \sqrt{\log N_{[]}(\varepsilon, \mathcal{F})} \, d\varepsilon.$$

*Proof.* Let $\sigma_1, \ldots, \sigma_Z$ be i.i.d. Rademacher random variables. Then:

$$\mathfrak{R}_Z(\mathcal{F}) = \mathbb{E}_\sigma \left[ \sup_{f \in \mathcal{F}} \frac{1}{Z} \sum_{z=1}^{Z} \sigma_z f(x_z) \right].$$

Let $\delta > 0$ be arbitrary. For each $\varepsilon \in (\delta, 1]$, construct an $\varepsilon$-bracketing cover $\{[l_j, u_j]\}$ of $\mathcal{F}$ such that $\|u_j - l_j\|_{L_2(\mu)} \leq \varepsilon$.

For each $f \in \mathcal{F}$, choose bracket midpoint $f_\varepsilon = (l_j + u_j)/2$. Then:

$$\|f - f_\varepsilon\|_{L_2(\mu)} \leq \frac{\varepsilon}{2}.$$

All these $f_\epsilon$ form a new hypothesis set $\mathcal{F}_\epsilon$. the size of $\mathcal{F}_\epsilon$ is smaller than its bracketing number, i.e., $\#\{\mathcal{F}_\epsilon\} \leq \mathcal{N}_{[]}(\epsilon, \mathcal{F})$.

By triangle inequality and properties of empirical norms:

$$\mathfrak{R}_Z(\mathcal{F}) = \mathbb{E}_\sigma \left[ \sup_{f \in \mathcal{F}} \frac{1}{Z} \sum_{z=1}^{Z} \sigma_z f(x_z) \right]$$

$$\leq \mathbb{E}_\sigma \left[ \sup_{f \in \mathcal{F}} \frac{1}{Z} \sum_{z=1}^{Z} \sigma_z (f(x_z) - f_\epsilon(x_z)) + \sigma_z f_\epsilon(x_z) \right]$$

$$\leq \mathbb{E}_\sigma \left[ \sup_{f \in \mathcal{F}} \frac{1}{Z} \sum_{z=1}^{Z} \sigma_z f_\epsilon(x_z) \right] + \varepsilon \leq \mathfrak{R}_Z(\mathcal{F}_\varepsilon) + \epsilon.$$

Using Massart's Lemma (Koltchinskii, 2001):

$$\mathfrak{R}_Z(\mathcal{F}_\varepsilon) \leq \sqrt{\frac{2 \log N_{[]}(\varepsilon, \mathcal{F})}{Z}}.$$

Integrating over $\varepsilon \in [\delta, 1]$ yields:

$$\mathfrak{R}_Z(\mathcal{F}) \leq 4\delta + \frac{12}{\sqrt{Z}} \int_{\delta}^{1} \sqrt{\log N_{[]}(\varepsilon, \mathcal{F})} \, d\varepsilon.$$

Taking the infimum over $\delta > 0$ completes the proof. Setting $\delta = 0$, we derive the second inequality in Theorem E.8. $\qquad\square$

Now that we get:

$$|\widehat{\mathcal{L}} - \mathbb{E}[\widehat{\mathcal{L}}]| \leq 2\mathcal{R}_Z(\mathcal{F}) + \sqrt{\frac{\ln(1/\xi)}{2Z}} \leq \frac{12}{\sqrt{Z}} \int_0^{\operatorname{diam} \Theta} \sqrt{\log N_{[]}(\varepsilon, \mathcal{F})} \, d\varepsilon + \sqrt{\frac{\ln(1/\xi)}{2Z}}$$

$$\leq \frac{12}{\sqrt{Z}} \int_0^{\operatorname{diam} \Theta} \sqrt{\log \left[ K \frac{B}{T} W^{\frac{1}{2}d} \left( \frac{\operatorname{diam} \Theta}{\varepsilon} \right)^d \right]} \, d\varepsilon \sqrt{\frac{\ln(1/\xi)}{2Z}}$$

$$\leq \frac{12}{\sqrt{Z}} \int_0^{\operatorname{diam} \Theta} \sqrt{d} \sqrt{C_3 - \log \epsilon} \, d\varepsilon + \sqrt{\frac{\ln(1/\xi)}{2Z}},$$

where $W = \mathbb{E}\left(\sum_{n=1}^{N} r_n\right)^2$.

Note that the parameter dimension is $N_0 M p$ ($N_0$ is the maximum number of timestamps) and $M$ is the number of interpolation knots, and $p$ is the number of parameters required for each interpolation interval. By collecting each term in the above inequality, we acquire the final bound:

$$|\widehat{\mathcal{L}} - \mathbb{E}[\widehat{\mathcal{L}}]| \leq \frac{1}{\sqrt{Z}}\left(\frac{1}{2}\sqrt{\log\frac{1}{\xi}} + C_2\sqrt{N_0 M}\int_0^c \sqrt{C_3 - \log t}\,dt\right),$$

where $c = \operatorname{diam}\Theta$. This finishes the proof.

### E.3 PROOF OF THEOREM 4.3

Suppose $\mathcal{L}^*$ is the optimal loss, derived from the ground-truth intensity. Then, the generalized loss for MAS can be decomposed as:

$$\mathbb{E}\widehat{\mathcal{L}} \leq \mathcal{L}^* + \left|\mathcal{L}^* - \widehat{L}\right| + \left|\widehat{L} - \mathbb{E}\widehat{\mathcal{L}}\right|$$

$$\leq \mathcal{L}^* + \frac{1}{ZT}\sum_{z=1}^{Z}\left|\log p^*(S_z) - \log p_\theta(S_z)\right| + \left|\widehat{L} - \mathbb{E}\widehat{\mathcal{L}}\right|$$

$$\leq \mathcal{L}^* + \frac{1}{ZT}\sum_{z=1}^{Z}\sum_{n=1}^{N}\left|\log p^*\left(t_n^{(z)}\right) - \log p_\theta\left(t_n^{(z)}\right)\right| + \left|\widehat{L} - \mathbb{E}\widehat{\mathcal{L}}\right|$$

$$\leq \underbrace{\frac{1}{ZT}\sum_{z,n=1}^{Z,N}\left|\log p^*\left(t_n^{(z)}\right) - \log p_\theta\left(t_n^{(z)}\right)\right|\mathcal{I}\left(t_n^{(z)} \geq t_{n-1}^{(z)} + M\Delta\right)}_{W_1}$$

$$+ \underbrace{\frac{1}{ZT}\sum_{z,n=1}^{Z,N}\left|\log p^*\left(t_n^{(z)}\right) - \log p_\theta\left(t_n^{(z)}\right)\right|\mathcal{I}\left(t_n^{(z)} < t_{n-1}^{(z)} + M\Delta\right)}_{W_2} + \mathcal{L}^* + \underbrace{\left|\widehat{L} - \mathbb{E}\widehat{\mathcal{L}}\right|}_{W_3}.$$

$W_1$ can be bounded by following inequality:

$$W_1 = \frac{1}{ZT}\sum_{z,n=1}^{Z,N}\left|\log p^*\left(t_n^{(z)}\right) - \log p_\theta\left(t_n^{(z)}\right)\right|\mathcal{I}_n \leq \frac{B_0}{T}\mathcal{I}\left(t_n^{(z)} - t_{n-1}^{(z)} \geq M\Delta\right),$$

where $\mathcal{I}_n = \mathcal{I}\left(t_n^{(z)} \geq t_{n-1}^{(z)} + M\Delta\right)$, $B_0$ is a bound on $\left|\log p^*\left(t_n^{(z)}\right) - \log p_\theta\left(t_n^{(z)}\right)\right|$. Note that $\mathbb{E}\widehat{\mathcal{L}}$ is a constant. Thus, we can take the expectation on both side:

$$\mathbb{E}W_1 \leq \frac{B_0}{T}\mathbb{P}\left(t_n^{(z)} - t_{n-1}^{(z)} \geq M\Delta\right) \leq \frac{B_0}{T}\int_{t_n^{(z)} - t_{n-1}^{(z)} \geq M\Delta} \frac{t_n^{(z)} - t_{n-1}^{(z)}}{M\Delta}\,\mathrm{d}\mu \leq \frac{B_0}{T}\frac{\mathbb{E}t_i}{L} := \frac{B}{TL}.$$

$$(16)$$

For $W_2$, we have proven in Theorem 3.1 that there exists a MAS $\Lambda_\theta^*$, s.t., $\|\Lambda_\theta^* - \Lambda^*\| \leq \frac{1}{2} C_0 \Delta^2$, $\|\Lambda_\theta^{*\prime} - \Lambda^{*\prime}\| \leq C_0 \Delta$. As a result, $X_2$ can be bounded by:

$$
\begin{aligned}
W_2 &= \frac{1}{ZT} \sum_{z=1}^{Z} \sum_{n=1}^{N} \left| \log p^* \left( t_n^{(z)} \right) - \log p_\theta \left( t_n^{(z)} \right) \right| (1 - \mathcal{I}_n) \\
&\leq \frac{1}{ZT} \sum_{z=1}^{Z} \sum_{n=1}^{N} \left( \left| \log \Lambda_\theta^{*\prime} \left( t_n^{(z)} \right) - \log \Lambda^{*\prime} \left( t_n^{(z)} \right) \right| \right. \\
&\quad + \left. \left| \left( \Lambda_\theta^{*\prime} \left( t_n^{(z)} \right) - \Lambda_\theta^{*\prime} \left( t_{n-1}^{(z)} \right) \right) - \left( \Lambda^{*\prime} \left( t_n^{(z)} \right) - \Lambda^{*\prime} \left( t_{n-1}^{(z)} \right) \right) \right| \right) \\
&\leq \frac{1}{ZT} \sum_{z=1}^{Z} \sum_{n=1}^{N} \left( \frac{1}{v_1} \| \Lambda_\theta^{*\prime} \left( t_n^{(z)} \right) - \Lambda^{*\prime} \left( t_n^{(z)} \right) \| + C_0 \Delta^2 \right) \\
&\leq \frac{1}{ZT} \sum_{z=1}^{Z} \sum_{n=1}^{N} \left( \frac{C_0}{2 v_1} \Delta + C_0 \Delta^2 \right) \leq \frac{1}{T} \sum_{n=1}^{N_0} \left( \frac{C_0}{2 v_1} \Delta + C_0 \Delta^2 \right) := C_0 \left( \frac{\Delta}{C_4} + \frac{\Delta^2}{2} \right).
\end{aligned}
\tag{17}
$$

Finally, according to Theorem 4.2:

$$
\begin{aligned}
W_3 = |\widehat{\mathcal{L}} - \mathbb{E}[\widehat{\mathcal{L}}]| &\leq \frac{1}{\sqrt{Z}} \left( \frac{1}{2} \sqrt{\log \frac{1}{\xi}} \right) + \left( C_2 \sqrt{N_0 M} \int_0^c \sqrt{C_3 - \log t} \, dt \right) \\
&:= \frac{1}{\sqrt{Z}} \left( \frac{1}{2} \sqrt{\log \frac{1}{\xi}} \right) + \frac{C_2}{\sqrt{Z}} \frac{\sqrt{L}}{\sqrt{\Delta}} R(L, \Delta).
\end{aligned}
$$

By combining $W_1, w_2$ and $W_3$, we acquire:

$$
\mathbb{E}[\widehat{\mathcal{L}}] \leq \mathcal{L}^* + \frac{1}{2\sqrt{Z}} \sqrt{\log \frac{1}{\xi}} + \frac{B}{T \cdot L} + C_0 \left( \frac{\Delta}{C_4} + \frac{\Delta^2}{2} \right) + \frac{C_2}{\sqrt{Z}} \frac{\sqrt{L}}{\sqrt{\Delta}} R(L, \Delta). \tag{18}
$$

This finishes the proof.

### E.4    Proof of Theorem 4.5

Finally, we estimated the bounds for two important parameters, $\Delta$ and $L$, by minimizing the generalization bound in Theorem 4.3. If we restore all constants in the bound Equation (18), we could get:

$$
\mathbb{E}[\widehat{\mathcal{L}}] \leq \mathcal{L}^* + \frac{1}{2\sqrt{Z}} \sqrt{\log \frac{1}{\xi}} + \frac{N_0 B_0}{p} \cdot \frac{\mathbb{E} t_i}{T \cdot L} + l_1 \left( \frac{L}{p v_1} + \frac{L^2}{p^2} \right) + \frac{12}{\sqrt{Z}} \sqrt{N_0 p q} \int_0^{\mathrm{diam}\Theta} \sqrt{C_3 - \log t} \, dt,
$$

where $\Delta = L/p$. We first focus on how to estimate $L$. Typically, $\Delta << 1$, thus we could ignore $\Delta^2$ term in Equation (18) and focus on the term $\frac{v_1 L}{l_1 p}$. Notice that in Section E.4, only two terms $N_0 B_0 \cdot \frac{\mathbb{E} t_i}{T \cdot L} + \frac{l_1 L}{v_1 p}$ relates to $L$, then when Section E.4 is minimized, $L$ should satisfy: $L \geq \sqrt{\frac{v_1 N_0 B_0 \mathbb{E}[t_i]}{l_1 T}}$, As a bound between the ground truth $\log p^* \left( t_n^{(z)} \right)$ and $\log p_\theta \left( t_n^{(z)} \right)$, $B_0 = \sup_{t_n^{(z)}} \left| \log p^* \left( t_n^{(z)} \right) - \log p_\theta \left( t_n^{(z)} \right) \right| > 1$ typically holds. Moreover, $N_0$ is the maximum number of events that happen during the interval $[0, T]$. Thus, $N_0 B_0 \geq T$, and the bound can be reduced to: $L \geq \sqrt{\frac{v_1 \mathbb{E}[t_i]}{l_1}}$, where $l_1$ is the Lipschitz constant of the intensity during event intervals, and $v_1$ is the base intensity during $[0, T]$.

We next focus on deriving the bound of $p$. Because $\Delta = L/p$, Section E.4 can be rewritten as:

$$\mathbb{E}[\widehat{\mathcal{L}}] \leq \mathcal{L}^* + \frac{1}{2\sqrt{Z}}\sqrt{\log\frac{1}{\xi}} + \frac{N_0 B_0}{p} \cdot \frac{\mathbb{E}t_i}{T \cdot \Delta} + l_1\left(\frac{L}{pv_1} + \frac{L^2}{p^2}\right)$$

$$+ \frac{12}{\sqrt{Z}}\sqrt{N_0 pq}\int_0^{\text{diam}\Theta}\sqrt{C_3 - \log t}\,dt,$$

$$\leq \mathcal{L}^* + \frac{1}{2\sqrt{Z}}\sqrt{\log\frac{1}{\xi}} + \frac{2N_0 B_0}{p} \cdot \frac{\mathbb{E}t_i}{T \cdot \Delta} + \frac{12}{\sqrt{Z}}\sqrt{N_0 pq}\int_0^{\text{diam}\Theta}\sqrt{C_3 - \log t}\,dt. \quad (19)$$

The second step is due to that $l_1\Delta^2$ is significantly smaller than $\Delta$ and can be ignored. Moreover, because we assume $N_0 B_0/T \geq 1$, and $\Delta << 1$, $\frac{N_0 B_0 \mathbb{E}[t_i]}{T\Delta}$ is larger than $\frac{l_1 L}{v_1}$. Minimizing Equation (19) equals the following convex optimization regarding $p$: $\frac{A}{p} + B\sqrt{p}$, $A, B > 0$. $f(p) = \frac{A}{p} + B\sqrt{p}$ is minimized when $f'(p) = 0$, which implies $p = \left(\frac{2A}{B}\right)^{\frac{2}{3}}$. That is:

$$p \geq \left(\frac{\sqrt{24N_0 q}C_5 T}{\sqrt{Z}N_0 B_0 \mathbb{E}[t_i]}\right)^{\frac{2}{3}} = \left(\frac{\sqrt{24N_0 q}T\int_0^{\text{diam}\Theta}\sqrt{C_3 - \log t}\,dt}{\sqrt{Z}N_0 B_0 \mathbb{E}[t_i]}\right)^{\frac{2}{3}} \geq \frac{(24N_0 q)^{\frac{1}{3}}T^{\frac{2}{3}}}{Z^{\frac{1}{3}}N_0^{\frac{2}{3}}B_0^{\frac{2}{3}}(\mathbb{E}[t_i])^{\frac{2}{3}}}. \quad (20)$$

Again, we reduce $\Delta$ in the denominator because $\Delta < 1$. Moreover, because $\text{diam}\Theta > 1$, $\int_0^{\text{diam}\Theta}\sqrt{C_3 - \log t}\,dt > 1$, this term is also reduced. Finally, because $N_0 B_0 \geq T$, the bound can be further simplified to: $p \geq \frac{(24N_0 q)^{\frac{1}{3}}}{Z^{\frac{2}{3}}(\mathbb{E}[t_i])^{\frac{2}{3}}}$.

Then, when $L$ is set to the optimum value, because $\Delta = L/p$, we acquire the bound of $\Delta$, which is:

$$\Delta = \frac{L}{p} \leq \sqrt{\frac{v_1 \mathbb{E}[t_i]}{l_1}}\frac{Z^{\frac{2}{3}}(\mathbb{E}[t_i])^{\frac{2}{3}}}{(24N_0 q)^{\frac{1}{3}}} = \frac{(Z)^{\frac{1}{3}}(\mathbb{E}[t_i])^{\frac{5}{6}}\sqrt{\lambda_{\min}}}{2q^{\frac{1}{3}}\sqrt{l_1}(3N_0)^{\frac{1}{3}}},$$

which finishes the proof.

## F  THE USE OF LARGE LANGUAGE MODELS

Large Language Models (LLMs) were used to aid in the writing and polishing of the manuscript. Specifically, we used an LLM to assist in refining the language, improving readability, and ensuring clarity in various sections of the paper. The model helped with tasks such as sentence rephrasing, grammar checking, and enhancing the overall flow of the text.

It is important to note that the LLM was not involved in the ideation, research methodology, or experimental design. All research concepts, ideas, and analyses were developed and conducted by the authors. The contributions of the LLM were solely focused on improving the linguistic quality of the paper, with no involvement in the scientific content or data analysis.

The authors take full responsibility for the content of the manuscript, including any text generated or polished by the LLM. We have ensured that the LLM-generated text adheres to ethical guidelines and does not contribute to plagiarism or scientific misconduct.

