# OpenReview forum: "Efficient Multivariate Temporal Point Process via Monotone Alternating Splines"
_ICLR.cc/2026/Conference — Submitted to ICLR 2026_

### Official Review · Reviewer_wrnA · 2025-10-30

**Soundness:** 4
**Presentation:** 4
**Contribution:** 4
**Rating:** 8
**Confidence:** 2

**Summary:**

The paper proposes **Monotone Alternating Splines**, a framework for parametrizing conditional cumulative intensity function of temporal point processes. The idea is to avoid numerical integration of the conditional intensity function during likelihood estimation, as this is computationally expensive and error prone. The conditional cumulative intensity is modelled in two parts: using monotone splines to fit the function within a defined interval (interpolation) and using a simple monotonic function (i.e. not a spline) to extrapolate outside this interval.

Unlike neural network-based methods, the use of splines means we get analytical derivatives, avoiding the need for automatic differentiation. Theoretical guarantees are offered for fitting and generalization, and empirical results on five synthetic and four real-world datasets indicate good performance.

**Strengths:**

The paper addresses a well-defined and important problem, namely the inefficiency and inflexibility of modelling TPPs. The proposed MAS framework appears to be an intuitive solution that tackles the limitations of CIF-based models and (cumulative) neural network models. The validation on nine datasets, including four real-world datasets, shows consistently competitive performance with all baselines. Standard deviations are reported for experimental results, adding some quantification of uncertainty to the claims.

The generalization analysis in Section 4 provides a clear, interpretable theoretical bound that decomposes the error into interpolation, extrapolation and complexity terms. This is nice to have for a practical method. The argument and evidence for MAS efficiency, thanks to its analytical derivatives, is convincing. The ablation study directly validates theoretical tradeoffs. The conceptual figures are quite clear and efficiently use the available space (assuming ICLR guidelines allow `wrapfig`).

**Weaknesses:**

The main criticism is in the presentation of some of the results. Figure 4(e) shows training time for what appears to be a single run, which is surely highly variable. The mean and standard deviation/error over multiple runs would be more convincing. Similarly, Figures 4(a-d) could benefit from some uncertainty quantification.

Furthermore, the scaling of timestamps by constant factors appears a bit _ad-hoc_. Is MAS sensitive to this scaling hyperparameter?

The paper's primary empirical evidence is presented in Tables 1 and 2, which are dense, difficult-to-read "leaderboard" tables. The use of bolding and underlining to highlight the top two models is a clear sign that the table format itself is failing to make the results accessible. The raw scores are not something that any reader is likely to need to look up directly, at least not in the main section of the paper. More rigorous and modern methods for comparing multiple models across multiple datasets are available, including the use of statistical tests (rather than reporting raw scores) and visual presentation, using small multiples or a critical difference diagram. See Demšar (2006; ["Statistical Comparisons of Classifiers over Multiple Data Sets." _JMLR_](http://jmlr.org/papers/v7/demsar06a.html)) or works by Edward Tufte.

### Minor points

1. In the introduction it is written _"asynchronous timestamps in continuous time, which sets them apart from conventional synchronized time series data..."_ This seems to be an odd use of the term "asynchronous" and "synchronous" when it would appear the authors mean really "irregular" and "regular", here.

2. The abbreviation "(TPPs)" is introduced in the abstract but never re-used.

3. The Figures use quite a modest amount of space and this is laudable but some of the labels and arrows are quite small, and only readable when zooming in, maybe some parts could be made slightly bigger.

**Questions:**

1. Could you please clarify how many runs were used to compute the means and standard deviations in Tables 1 and 2?

2. For Figure 4(e), could you provide the mean and standard deviation for the training times over multiple runs?

3. The data scaling in the appendix (Section D.2) seems necessary to get MNN-based models to converge. Did you test if MAS is also sensitive to this scaling, or does it converge robustly even without it?

---

> ### Author Response · Authors · 2025-11-20
> **Response to Reviewer wrnA**
>
> **Q1: Figure 4(e) shows training time for what appears to be a single run, which is surely highly variable. The mean and standard deviation/error over multiple runs would be more convincing.**
>
> A: Thank you for your comments! In the updated Figure 4(e), we added runtime comparisons with more baselines (including RMTPP and THP). In the appendix, we also report the mean and standard deviation of these baselines over three repeated runs under the same settings at both 10 and 50 epochs.
>
> ---
>
> **Q2: The scaling of timestamps by constant factors appears a bit ad-hoc. Is MAS sensitive to this scaling hyperparameter? Did you test if MAS is also sensitive to this scaling, or does it converge robustly even without it?**
>
> A: Thank you for your comments! For each dataset, we followed the scaling factors used in [1], where the same scaling is applied in their public GitHub implementation. Our main motivation for adopting these scaling factors is to accommodate CCIF methods based on MNNs. As the depth of an MNN increases, each layer produces a monotonic output, and a large $t$ can cause $\Lambda(t)$ to grow exponentially. MAS does not use an MNN-based architecture and therefore does not suffer from this issue. Based on our experiments, adding or removing this scaling factor has no noticeable effect on the convergence speed or final performance of MAS.
>
> ---
>
> **Q3: In the introduction it is written "asynchronous timestamps in continuous time…" This seems to be an odd use of the term "asynchronous" and "synchronous" and the authors likely mean “irregular’’ and “regular’’ here.**
>
> A: Thank you for pointing this out! We have corrected this in the revised manuscript.
>
> ---
>
> **Q4: The abbreviation "(TPPs)" is introduced in the abstract but never re-used.**
>
> A: Thank you for pointing this out! We have replaced many occurrences of “point process’’ in the main text with “TPPs’’.
>
> ---
>
> **Q5: The figures use a modest amount of space but some labels and arrows are small and only readable when zooming in.**
>
> A: Thank you for pointing this out! We have enlarged the font sizes in Figures 2 and 3 as much as possible. In the revised Figure 4(e), we also increased the legend font size.
>
> ---
>
> **Q6: Could you please clarify how many runs were used to compute the means and standard deviations in Tables 1 and 2?**
>
> A: Thank you for your question! All experiments were repeated three times, and we report the mean and standard deviation.
>
> ---
>
> **References**
>
> [1] Haoqun Cao, et al. Is Score Matching Suitable for Estimating Point Processes?

---

> > ### Comment · Reviewer_wrnA · 2025-11-25
> >
> > Thanks to the authors for carefully addressing the points raised. My score remains the same (i.e. accept).

---

> > > ### Author Response · Authors · 2025-11-26
> > >
> > > Thank you very much for your thoughtful follow-up and your positive assessment of our work. We truly appreciate your recognition and are greatly encouraged by your supportive feedback.

---

### Official Review · Reviewer_mUbi · 2025-10-31

**Soundness:** 2
**Presentation:** 3
**Contribution:** 1
**Rating:** 2
**Confidence:** 3

**Summary:**

This paper introduces Monotone Alternating Splines (MAS), a new method for efficiently modeling multivariate temporal point processes through the conditional cumulative intensity function (CCIF). Unlike conventional methods via monotone neural networks, MAS combines spline-based interpolation and simple monotone extrapolation, ensuring both flexibility and computational efficiency. The interpolation enhances fitting ability, while the extrapolation guarantees generalization. Experiments on both synthetic and real-world datasets demonstrate that MAS outperforms existing methods in terms of predictive performance.

**Strengths:**

- The manuscript is clearly written and easy to follow.
- The result demonstrating higher accuracy than other major models is impressive from a practical standpoint.

**Weaknesses:**

- The primary concern lies in the technical contribution’s weakness. The idea of modeling the cumulative intensity function with a monotone spline itself is not particularly novel [1]. While separating the modeling of interpolation and extrapolation is practically useful, it remains a minor methodological refinement. Also, representing the cumulative intensity function with a monotone spline is mathematically equivalent to representing the intensity function with a nonnegativity-constrained spline. Since the latter approach has already been explored (e.g., [2]), it is unclear what additional significance the proposed idea of modeling the cumulative intensity function using a monotone spline truly provides.
- If the paper’s main contribution lies in its empirical accuracy (Tables 1 and 2), the experimental setup requires further improvement. For example, in fitting a Hawkes process with a simple triggering kernel and limited data, simpler models naturally achieve higher accuracy. Neural network–based models tend to underperform when the dataset is small unless their architecture (layer depth, unit size, and especially the smoothness of activation functions) is carefully tuned to the dataset. To fairly demonstrate that the proposed model outperforms existing methods in terms of both flexibility and generality, the baseline monotone NN architectures should be determined via cross-validation, and the comparison should be conducted across multiple data sizes.

[1] Bantis et al., "Survival estimation through the cumulative hazard function with monotone natural cubic splines", Lifetime data analysis, 2012.

[2] Loaiza-Ganem et al. "Deep random splines for point process intensity estimation of neural population data", NeurIPS, 2019.

**Questions:**

- The explanation of the synthetic data is insufficient. For the Hawkes process datasets, over what time span was the data generated? What was the scale of the dataset used for training?
- The experimental results only report accuracy. Since the paper claims efficiency as a key advantage of the proposed model, the training time should also be provided.
- Why is the second condition in Theorem 3.1 necessary? The explanation refers to the Hawkes process, but by adopting a jump-type triggering kernel, one can construct a Hawkes process that does not satisfy this condition. Therefore, this second condition may not be a general requirement for cumulative intensity functions.
- Theorem 3.1 is presented as self-evident, but references should be provided for its proof.
- The paper states that MNNs lack expressive power (lines 180-181), but the reason or evidence for this claim should be explained.

---

> ### Author Response · Authors · 2025-11-20
> **Response to Reviewer mUbi (Part 1 of 2, Question 1-3)**
>
> **Q1: It is unclear what additional significance the proposed idea of modeling the cumulative intensity function using a monotone spline truly provide.**
>
> A:
> Thank you for your response. Although many prior TPP models have used ideas related to monotone splines, we believe that MAS introduces innovations both **technically** and **theoretically** compared with these approaches.
>
> At the technical level, we believe that MAS introduces innovations compared with both prior spline-based approaches and prior CCIF-modeling approaches.
>
> - **First, MAS is simpler to train and more efficient than other spline based methods.**
>
>     - While many existing methods also adopt RQS or other spline parameterizations, they typically model variables other than the CCIF (e.g., the intensity function [1] or quantile function [2]), which prevents them from being trained directly with maximum likelihood. For example, [1] uses deep random splines, but the conditional intensity is governed by latent variables, making the likelihood intractable and forcing the use of variational inference.  **Because these approaches cannot use exact MLE, they often suffer from weaker optimization signals, slower convergence, and higher computational overhead compared with likelihood-based training.**
>
>     - In contrast, MAS models the CCIF directly using monotone splines combined with extrapolation functions. This choice of intermediate representation **makes the likelihood fully tractable**, enabling exact MLE training and significantly improving computational efficiency.
>
>
> - **Second, MAS adopts a broader class of spline types and explicitly accounts for CCIF's tail behavior.**
>
>     - On the one hand, if one directly models the CIF using splines, many spline families (except M-splines and B-splines) do not guarantee non-negativity. As a result, methods that spline-model the CIF are only constrained to a limited subset of spline bases, whereas **MAS can employ a wide range of interpolation functions** such as RQS, cubic splines, RLS, and more.
>
>     - On the other hand, **MAS explicitly incorporates the effect of tail functions. MAS can employ different tail functions to model the CCIF, allowing it to adapt to different data characteristics**. For example, a linear tail fits Poisson-like processes, while an exponential tail fits Hawkes-like processes. Our additional ablation experiments further demonstrate that the choice of tail distribution improves model performance, which is an aspect that previous spline-based CIF modeling methods did not consider.
>
> - **Thirdly, compared to other CCIF-based methods, MAS is more flexible and breaks the long-standing convex trap of MNNs.**
>
>     - It is important to emphasize that mainstream positive-weighted MLPs used for CCIF modeling, including [3] [4] [5], have **inherent theoretical limitations** in expressive power: **they can only represent convex CCIFs**. Moreover, this limitation is deeply rooted in the structure of positive-weighted MLPs and is difficult to overcome (we discuss this thoroughly in Question 7).
>
>     - MAS not only removes this constraint and provides theoretical approximation guarantees, but also further improves computational efficiency, since it avoids the double backpropagation required by MNN-based methods.
>
> - **Finally, our method provides theoretical advances.** Unlike prior CCIF-based models, our work analyzes both the approximation error and the generalization performance of MAS. By establishing generalization bounds, we further derive principled guidelines for selecting model hyperparameters such as the total interpolation length and the interpolation interval size (**Corollary 4.5**). To the best of our knowledge, previous CCIF-based approaches focus primarily on empirical improvements and do not provide a theoretical analysis of CCIF-oriented models. MAS fills this theoretical gap.
>
> ---
>
> **Q2: The explanation of the synthetic data is insufficient. For the Hawkes process datasets, over what time span was the data generated? What was the scale of the dataset used for training?**
>
> A: Thank you for the clarification. For the synthetic datasets, we follow exactly the same configuration as used in [5]. Specifically, we sampled 1,000 sequences over the interval $[0,100]$, with 60\% used for training, 20\% for testing, and the remaining 20\% for validation.
>
> ---
>
> **Q3: The experimental results only report accuracy. Since the paper claims efficiency as a key advantage of the proposed model, the training time should also be provided.**
>
> A: Thank you for your correction. In **Figure 4(e)**, we present a comparison of the efficiency of MAS with THP, RMTPP, and other CCIF-based methods. We have added detailed experiment results in **Table 7** to further demonstrate the computational efficiency of MAS.
>
> **Please see Part 2 for responses to the remaining questions and citations**

---

> ### Author Response · Authors · 2025-11-20
> **Response to Reviewer mUbi (Part 2 of 3, Question 4-6)**
>
> **Continued from Part 1**
>
> ---
>
> **Q4: To fairly demonstrate that the proposed model outperforms existing methods in terms of both flexibility and generality, the baseline monotone NN architectures should be determined via cross-validation, and the comparison should be conducted across multiple data sizes.**
>
> A: Thank you for the suggestion.
> To rigorously demonstrate that MAS indeed outperforms other baselines, we conduct a cross-validation study comparing MAS with a classical CCIF-based method, FullyNN, which utilizes MNN to model CCIF. **We evaluate FullyNN under different network depths, widths, activation functions, and training data sizes to ensure that MAS’s superiority is not merely due to accidental hyperparameter choices.**
>
> Specifically, we fix the encoder architecture of MAS and FullyNN to be identical (i.e., both models receive the same input features). Following the data generation procedure of the hawkes1 dataset, we independently sample 400 and 800 sequences within the time range $[0,100]$ for training. During training, we vary FullyNN’s number of fully connected layers from 1 to 3, its hidden widths between 16 and 32, and its activation functions between softplus and sigmoid (representing different levels of smoothness). All Training strategy are the same as the main experiement. We then compare its NLL and RMSE with those of MAS, as is presented in **Table 6** in the appendix.
>
> The results show that when the network depth exceeds two layers, the performance of FullyNN no longer improves significantly with additional depth. **Moreover, regardless of whether softplus or sigmoid is used, MAS consistently outperforms FullyNN across all configurations.** This cross-validation experiment further confirms the advantage of MAS over classical CCIF-based modeling methods.
>
> ---
>
> **Q5: Why is the second condition in Theorem 3.1 necessary? The explanation refers to the Hawkes process, but by adopting a jump-type triggering kernel, one can construct a Hawkes process that does not satisfy this condition. Therefore, this second condition may not be a general requirement for cumulative intensity functions.**
>
> A: Thank you for your comment. The expression here was indeed not rigorous. When we stated the requirements for CCIF, we were considering commonly used Hawkes processes, such as those with exponential kernels, sinusoidal kernels, and self-correcting processes. And these classic TPPs all suit these conditions. We agree that presenting 3.1 as a standalone theorem is not appropriate. In the revised version of the paper, **we no longer refer to it as a theorem and explicitly clarify that these conditions only describe common classical TPPs.**
>
> ---
>
> **Q6: The experimental results only report accuracy. Since the paper claims efficiency as a key advantage of the proposed model, the training time should also be provided.**
>
> A: Thank you for your correction. In **Figure 4(e)**, we present a comparison of the efficiency of MAS with THP, RMTPP, and other CCIF-based methods. We have added detailed experiment results in **Table 7** to further demonstrate the computational efficiency of MAS.
>
> ---
>
> **Please see Part 3 for responses to the remaining question and citations**

---

> > ### Author Response · Authors · 2025-11-20
> > **Response to Reviewer mUbi (Part 3 of 3, Question 7)**
> >
> > **Continued from Part 2**
> >
> > ---
> >
> > **Q7: The paper states that MNNs lack expressive power (lines 180-181), but the reason or evidence for this claim should be explained.**
> >
> > A: Thank you for your advice. Existing CCIF-based methods [3] [4] [5] typically utilize a positive-weighted MLP to model CCIF. However, this parametrization has severe limitations.
> >
> > The core drawback is that such MNNs are **inherently convex**. Let $\boldsymbol{h}^{(l)}$ denote the hidden state at layer $l$, the forward propagation of an $L$-layer positive-weighted MLP can be defined as:
> >
> > $$\Lambda(t) =\textbf{W}^{(L)} \sigma ( \cdots\sigma( \textbf{W}^{(2)} \sigma ( \textbf{W}^{(1)} t + \textbf{b}^{(1)} )+ \textbf{b}^{(2)}))+\textbf{b}^{(L)},$$
> >
> > where $\sigma(\cdot)$ is the activation function, and $\textbf{W}^{(l)}$ is the model weights. Its derivative regarding $t$ is:
> >
> > $$\lambda(t)  = \textbf{W}^{(L)} ( \prod_{l=L-1}^{1} \text{diag}[ \sigma'(\textbf{z}^{(l)}) ] \textbf{W}^{(l)} ).$$
> >
> > Since common activation functions (e.g., ReLU, Softplus) are non-decreasing, their derivatives satisfy $\sigma'(\cdot) \ge 0$. Thus, by enforcing the MLP parameter
> > $\boldsymbol{W}^{(l)} \ge 0$, the intensity function $\lambda(t)$ modeled by this MNN remains positive.
> >
> > However, MNNs typically use activation functions such as ReLU or softplus; however, these functions are **usually convex**. Their positive linear combinations in an MNN are therefore also **convex** with respect to
> > $t$. As a result, the CCIF modeled by such MNN-based approaches is **inherently convex**, making it unable to capture more flexible CCIF shapes.
> >
> > For example, in an exponentially decaying Hawkes process, the intensity function is monotonically decreasing. In other words, the second derivative of the CCIF is negative $\Lambda''(\cdot)<0$, so the CCIF is non-convex. In this case, **an MNN cannot properly model the final decay behavior of the Hawkes process**.
> >
> > Meanwhile, MNN methods based on this scheme also have other limitations. For example,
> >
> > - as the network depth increases, they are prone to numerical explosion;
> > - their optimization tends to be slow;
> > - when computing gradients (especially when using ReLU as the activation function), gradient vanishing can occur.
> >
> > On the other hand, moving beyond the “positive-weighted MLP” framework and designing a new MNN architecture capable of modeling the CCIF is extremely challenging. To the best of our knowledge, no existing work has proposed an MNN-based CCIF-modeling approach outside of the positive-weighted MLP paradigm. Precisely because of these difficulties, we propose a new MAS-based scheme for modeling the CCIF.
> >
> > **Reference**
> >
> >
> > [1] Gabriel Loaiza-Ganem, et al, Deep Random Splines for Point Process Intensity Estimation of Neural Population Data.
> >
> > [2] Souhaib Ben Taieb, Learning Quantile Functions for Temporal Point Processes with Recurrent Neural Splines.
> >
> > [3] Maolin Wang, et al, Cumulative Distribution Function-based General Temporal Point Processes.
> >
> > [4] BingqingLiu, Cumulative Hazard Function Based Efficient Multivariate Temporal Point Process Learning.
> >
> > [5] Takahiro Omi, Fully Neural Network Based Model for General Temporal Point Processes.

---

> ### Author Response · Authors · 2025-11-26
>
> Dear Reviewer mUbi,
>
> If we understand correctly, your main concerns relate to the novelty of our approach, the completeness of the experimental setup, and several clarifications regarding data generation, efficiency results, theoretical conditions, and the expressive limits of MNNs.
>
> In our response, we addressed these points by explaining the technical and theoretical contributions of MAS, adding further experiments and efficiency comparisons, providing missing dataset details, refining our theoretical statements, and elaborating on why classical MNNs are inherently limited.
>
> We kindly ask you to review our replies and let us know whether they resolve your concerns. Please feel free to let us know if anything remains unclear. We sincerely appreciate your time and feedback.

---

### Official Review · Reviewer_SkWr · 2025-11-01

**Soundness:** 2
**Presentation:** 2
**Contribution:** 2
**Rating:** 4
**Confidence:** 2

**Summary:**

The work proposed to use monotone alternating splines (MAS) to parameterize the cumulative conditional intensity function for modelling temporal point process. The MAS consisting of interpolation parts and extrapolation parts. This approach does not rely on numerical integration and does not have the problems of low computational efficiency and numerical approximation error. Theoretical study shows the approaches ability to fit complex TPPs and generalize effectively. Empirical study also shows competitive performance of the approach against a series of baselines on fitting TPP sequences and next event forecasting.

**Strengths:**

1. The work is well motivated, aiming at tackling some important and significant problems in TPP modelling, including the computational efficiency and approximation error.
2. Experiment results on next event predictions and log likelihood evaluation demonstrates the strong performance of the work against the baselines.

**Weaknesses:**

1. Despite the solid motivations, the innovation of the work can be summarized as a new approach toward parameterize the cumulative intensity function. This innovation itself is incremental.
2. Even improved efficiency is a central contribution of the work, good quantitative performance on TPP benchmarks are also critical for a TPP model. Many recent works on temporal point process evaluate TPP models on multiple-events long-horizon predictions [1]. Lacking such experiment results makes the empirical study of the work relatively weaker.
3. There are some recent SOTA TPP models not compared against in the model, including C-Diff [2], Hypro [3], Decomposable TPP [4]. The work should include these models in the comparison.
4. The ablation study results shows that the model performance is easily impacted by the selection of hyper-parameters including, interpolation length, number of knots, and interpolation function. However, the selection of these hyper-parameters are purely empirical without principled guidance. The existence of these hyper-parameters also makes the contributions of the work less significant.
5. The work claims that one of the limitations of existing intensity-based TPP approaches is the approximation error due to Monte-Carlo integration. This claim is not fully substantiated. In other words, can the performance of CIF-based models be improved and close the gap with the proposed approach with more accurate numerical integration?

References

[1] Xue, S. EasyTPP: Towards Open Benchmarking Temporal Point Processes, ICLR 2024

[2] Zeng, Mai. Interacting diffusion processes for event sequence forecasting. ICML 2024.

 [3] Xue, Siqiao, et al. "Hypro: A hybridly normalized probabilistic model for long-horizon prediction of event sequences." NeurIPS 2022

[4] Panos, Aristeidis. "Decomposable Transformer point processes." NeurIPS 2024.

**Questions:**

1. Is the same set of hyper-parameters like the number of interpolation nots or the interpolation length applied to all datasets or different hyper-parameters are selected for different datasets? Is it possible to dynamically change the hyper-parameters based on the local context of the event sequence? For example, should we have shorter interpolation lengths or more interpolation nots where event occurrences are denser?
2. If the next event happens at a very distant future, is the conditional NLL likelihood evaluation or the forecasting of that distant future events going to be overly reliant on the extrapolation function?

---

> ### Author Response · Authors · 2025-11-20
> **Response to Reviewer SkWr (Part 1 of 2, Question 1-3)**
>
> **Q1: Experiments on multiple-events long-horizon predictions should be included.**
>
> A:  Thank you for your advice. We conduct long-horizon prediction experiments on a univariate dataset
> (Hawkes1) and a multivariate dataset (TAXI). We compare MAS against THP and two CCIF multi-variate modeling methods (CuFun and EMTPP). For each model, we require it to recursively predict the next 3/5/10 events. The results are shown in the **Table 3**. As can be seen, MAS not only achieves high accuracy in single-step prediction but also maintains its superiority in long-horizon prediction.
>
> ---
>
> **Q2: Include C-Diff, HyPro, and Decomposable TPP as baselines.**
>
> A: Thank you for the recommendation. We have added HyPro and Decomposable TPP to our experimental evaluation on both univariate and multivariate datasets (shown in **Table1 and Table2**). However, we did not include the results of C-Diff in the tables because some metrics (such as NLL) cannot be computed. We found that C-Diff is a diffusion-model–based approach for modeling multivariate TPPs, and it does not directly model the NLL of the TPP. Due to the absence of these metrics, we did not compare it against MAS.
>
> ---
>
> **Q3: The ablation study results show that the model performance is easily impacted by the selection of hyperparameters including interpolation length, number of knots, and interpolation function. However, the selection of these hyperparameters is purely empirical without principled guidance.**
>
> A: Thank you for the comment. We indeed observe that MAS's test performance can vary with several hyperparameter choices. Nevertheless, even accounting for these variations, MAS still achieves substantially better results than the other baselines in the ablation study, which we believe demonstrates its empirical advantage.
>
> Regarding hyperparameter selection, at the end of the **Generalization Analysis** section we provide a detailed discussion of how to choose $\Delta$ and $L$.
> **Specifically, we provide a new theorem, Corollary 4.5 in the revised version**: When the generalization bound is minimized, the total interpolation length $L$ and interpolation interval $\Delta$ satisfy:
>
> $$L\ge \sqrt{\frac{\mathbb{E}[t_i] \cdot \lambda_{\text{min}}}{l_1}}, \Delta\le\frac{(Z)^{\frac{1}{3}}(\mathbb E [t_i])^{\frac{5}{6}}\sqrt{\lambda_{\text{min}}}}{\sqrt{l_1}q^{\frac{1}{3}}(24N_0)^{\frac{1}{3}}}.$$
>
> where $\mathbb E[t_i]$ presents the expectation of the event interval, $q$ is the freedom of each interpolation knot, $\lambda_{\text{min}}$ is the base intensity during $[0,T]$, and $l_1$ is the Lipschitz constant of the intensity during event intervals.
>
>
>
> According to **Corollary 4.5**, hyperparameters $L$ and $\Delta$ depend on two dataset-specific quantities:
>
> - the expected inter-event time $\mathbb{E}[t_i]$,
> - the degree of temporal variability in the TPP intensity, captured by its Lipschitz constant $l_1$.
>
> As $\mathbb{E}[t_i]$ grows, a longer interpolation horizon, i.e., a larger $L$, is required to track changes in the CCIF accurately.
> Moreover, as the intensity fluctuation, inflected by $l_1$, becomes sharper, MAS should provide a finer-grained approximation, which requires a smaller interpolation $\Delta$.
>
> Furthermore, **Corollary 4.5** provides a simple plug-in strategy for selecting $L$ and $\Delta$:
> For synthetic datasets where the functional form of the intensity is known, $l_1$, $\lambda_{\min}$, and $\mathbb{E}[t_i]$ can be estimated directly from samples, yielding an approximate lower bound for $L$ and $\Delta$.
> For general real-world datasets, inspired by Scott’s rule, we suggest assuming that the data follow a Hawkes process and estimating $L$ accordingly.
> The detailed procedure is provided in the appendix.
> Given the estimated $L$ and $\Delta$, the number of interpolation knots $p$ can also be calculated as $p=L/\Delta$.
>
> It is worth noting that, although an excessively large $L$ and an overly small $\Delta$ may theoretically lead to overfitting and thus increased generalization error, such overfitting can be effectively mitigated by standard training techniques (e.g., dropout [1] or SGD-based optimization [2]). In practice, we observe that this overfitting effect is mild.
> Therefore, we recommend using a moderately large $L$ and a relatively fine $\Delta$, consistent with the bounds provided in **Corollary 4.5**.
>
> **Reference**
>
> [1] Nitish Srivastava, et al, Dropout: A Simple Way to Prevent Neural Networks from Overfitting.
>
> [2] Nitish Shirish Keskar, et al, Improving Generalization Performance by Switching from Adam to SGD.
>
> ---
>
> **Please see Part 2 for responses to the remaining questions (approximation error due to numerical integral, hyper-parameters, and event prediction in the distant future).**

---

> > ### Author Response · Authors · 2025-11-20
> > **Response to Reviewer SkWr (Part 2 of 2, Question 4-6)**
> >
> > **Continued from Part 1.**
> >
> > ---
> >
> > **Q4: The work claims that one of the limitations of existing intensity-based TPP approaches is the approximation error due to Monte-Carlo integration. This claim is not fully substantiated. In other words, can the performance of CIF-based models be improved and close the gap with the proposed approach with more accurate numerical integration?**
> >
> > A:
> > Thank you for your suggestion! In our view, methods that model the CIF are generally affected by numerical integration, and **this influence cannot be completely eliminated simply by using a “more accurate” integration scheme**. Numerical integration not only introduces estimation errors but also reduces computational efficiency. This is because numerical integration—whether using quadrature or Monte Carlo methods—requires sampling on the time axis and performing additional forward passes to compute intensity values. Compared with a single-step forward pass, this is inherently less efficient. Moreover, using more accurate numerical integration (finer sampling or more complex integration estimators) further increases computational cost.
> >
> > We believe that MAS has an advantage in that **it computes exact CIF and CCIF in a single forward process**. This means MAS requires neither the multiple forward passes needed by CIF-based methods for integration, nor the two-step autograd used in MNN-based methods. **As a result, MAS enjoys advantages in both accuracy and efficiency**. Even if CIF-based methods adopt more refined numerical integration techniques, MAS's advantages cannot be completely eliminated.
> >
> >
> > ---
> >
> > **Q5: Is the same set of hyper-parameters---such as the number of interpolation knots or the interpolation length---applied to all datasets, or are different hyper-parameters selected for different datasets? Is it possible to dynamically change the hyper-parameters based on the local context of the event sequence?**
> >
> > A: Thank you for the question. Although, for fairness, we used the same set of hyper-parameters across all datasets in the experimental section, **the design of MAS indeed allows each interpolation interval length to be learned**. After an event $t_i$ occurs, the historical representation $\textbf{h}$ is updated, and this representation is then decoded into the MAS parameters. As a result, each interpolation interval length can be determined by this decoded parameter. In other words, the neural network can adaptively determine the interpolation interval lengths based on the local context of the event sequence. Similarly, the total interpolation length can also be learned by the network.
> >
> > ---
> >
> > **Q6: If the next event happens in the very distant future, will the conditional NLL evaluation or the forecasting of that event rely excessively on the extrapolation function?**
> >
> > A: Thank you for the comment. Indeed, if an event occurs far in the future, the extrapolation region contributes more heavily, while the interpolation region plays a smaller role. **MAS incorporates two mechanisms to mitigate this issue**:
> >
> > - **When selecting the hyper-parameter $L$, use the bound introduced in Corollary 4.5**. This bound is positively correlated with the average inter-event time. If many events in the sequence occur at very distant future timestamps, the bound correspondingly increases, requiring a larger total interpolation length to minimize generalization error, which helps alleviate the problem.
> >
> > - **Choosing an appropriate extrapolation function.** If the extrapolation function can already capture the behavior of the underlying TPP, then events falling into the extrapolation region will not degrade the fitting accuracy. For example, if the dataset exhibits Hawkes-like dynamics, i.e., self-excitation followed by gradual decay, an exponential tail is more suitable; conversely, if the data behaves more like a Poisson process with intensity stabilizing to a constant value, a linear tail is preferable. A well-chosen extrapolation function can therefore effectively address this issue.

---

> ### Author Response · Authors · 2025-11-26
>
> Dear Reviewer SkWr,
>
> If we understand correctly, your main concerns relate to: (1) the need to include long-horizon multi-event prediction experiments and additional baseline models, (2) the principled selection of hyperparameters, (3) the role of numerical integration in CIF-based approaches, (4) clarification on whether hyperparameters can adapt to local event contexts, and (5) the reliance on extrapolation when predicting far-future events.
>
> In our response, we addressed these points by adding long-horizon experiments and additional baselines, providing theoretical guidance for hyperparameter selection through Corollary 4.5, explaining why numerical integration inherently limits CIF-based methods, clarifying MAS’s ability to adapt hyperparameters dynamically through the decoder, and discussing how interpolation length and tail-function choices mitigate long-horizon extrapolation issues.
>
> We kindly ask you to review our replies and let us know whether they resolve your concerns. Please feel free to contact us if anything remains unclear. We sincerely appreciate your time and constructive feedback.

---

> ### Comment · Reviewer_SkWr · 2025-11-28
> **Post-Rebuttal Acknowledgement**
>
> I would like to thank the authors for addressing my concerns. Most of them are resolved but I have some questions specific about the new experiment results. It seems that the new results on the baseline model DTPP does not match the results in the original paper. Can the author explain this difference? It is also weird that the author compares against HYPRO in next event prediction experiments but not long-horizon forecasting as HYPRO is a strong baseline for long-horizon prediction.

---

> > ### Author Response · Authors · 2025-11-28
> > **Response to Reviewer SkWr (Questions about new experiment results)**
> >
> > Thank you for your valuable feedback! Regarding the experimental results, we would like to offer the following clarifications:
> >
> > **Regarding Decomposable TPP (DTPP)**: We identified a critical implementation error in the official code provided by the author of DTPP. Specifically, in the calculation of the RMSE metric, the final metric in DTPP paper is not square-rooted (In DTPP's GitHub repository, this bug was reported by someone else a few months ago, although the authors have not responded yet). Consequently, the values reported in their paper were effectively Mean Squared Error (MSE), not RMSE. We suspect this is the primary reason why some results reported in that paper appeared surprisingly low. After correcting this bug and re-evaluating DTPP—while strictly controlling for variables such as the Transformer backbone and data preprocessing—we obtained results that align with those presented in our submission.
> >
> > **Regarding HyPro**: We would like to clarify that HyPro is not a model architecture, but rather a training and inference paradigm specifically designed to mitigate cumulative errors in long-horizon prediction. In its original formulation, HyPro introduces an energy function on top of backbone models such as NHP or AttNHP to improve their long-horizon predictive performance. Since our proposed MAS is fundamentally an autoregressive TPP model, it is naturally compatible with the HyPro framework.
> >
> > In our long-horizon experiments, our primary goal is to compare different backbone architectures. MAS exhibits lower single-step and cumulative errors compared with traditional autoregressive baselines (as shown in Tables 1,2,3), demonstrating its effectiveness as a stronger backbone model. In principle, any compatible backbone, including MAS, could further benefit from the HyPro paradigm to achieve additional improvements in long-horizon prediction.

---

### Official Review · Reviewer_TZnp · 2025-11-01

**Soundness:** 2
**Presentation:** 2
**Contribution:** 2
**Rating:** 4
**Confidence:** 4

**Summary:**

The paper proposes **MAS** (Monotone Alternating Splines) for multivariate temporal point processes. It parameterizes the **conditional cumulative intensity function (CCIF)** after each event via a piecewise monotone spline for interpolation on $((t_n, t_n+L])$ and a simple extrapolator (linear or exponential) on $((t_n+L,\infty))$. Formally,  This decouples local flexibility (splines) from global monotonicity (tail), aiming to avoid architectural constraints of monotone neural nets (MNN).

Theory: (i) an interpolation error bound for the spline part; (ii) a generalization bound decomposing probability, extrapolation, and complexity terms; (iii) representability of Poisson/Hawkes. Experiments compare against CIF-based (RMTPP/THP/SAHP), CCIF/MNN (FullyNN/EMTPP), TriTPP (spline/flow), and intensity-free/score-matching baselines, reporting competitive or better NLL/RMSE on several datasets.

**Strengths:**

- **Clean parameterization**: Alternating “spline (local) + tail (global)” design for CCIF is intuitive, implementable, and potentially more flexible than weight-constrained MNNs.
- **Theory present**: Interpolation error and a three-term generalization bound provide a lens to discuss trade-offs (($\Delta, L, Z$)).
- **Empirical competitiveness**: Broad set of baselines and datasets; MAS is often competitive and occasionally best.
- **Engineering practicality**: The decoupling offers a reasonable path to stability and monotonicity in multivariate TPPs.

**Weaknesses:**

1. **Related-work inaccuracies / mispositioning**
- States/leans that **TriTPP only suits NHPP**; this conflicts with the **time-rescaling theorem** (general TPPs with integrable intensity admit such mappings). The issue harms credibility of the positioning.
- Overgeneralizes that “most monotone splines don’t preserve monotonicity outside the interpolation interval.” In practice, **RQS with linear tails** (as used in neural spline flows and TriTPP variants) ensures global monotonicity; the paper should contrast **its tail choice** (e.g., exponential) as a design tuned to TPPs, not as if splines lack tails.

2. **Formula/technical errors**
- **Hawkes CCIF corollary** uses $(t_n)$ in every exponential term; it should sum over all past events (t_i) with $(\exp(-\beta (t - t_i)))$. This is a visible typo in a central sanity-check.

3. **Completeness**
- Several wins are within one standard deviation;
- Missing key baselines: contemporary CDF/CCIF (e.g., CuFun/UMNN-style monotone flows) and parametric MLE Hawkes on synthetic data. Without them, it’s hard to assess where MAS truly dominates.

4. **Hyperparameters/fairness**
- MAS introduces extra knots (p) and window (L). Although defaults are given (e.g., (p=10, L=6)) and some ablations exist, the selection guidance from the bound is not operational (unknown constants), and per-dataset sensitivity isn’t systematically reported.
- Some results show higher variance for MAS than baselines, hinting at stability/tuning issues.

**Questions:**

1. Please correct the Hawkes corollary and re-run any sanity checks that depend on it.
2. Can you reposition TriTPP under the time-rescaling lens and explain, with evidence, when MAS’s exponential tail outperforms RQS linear tails (e.g., long-tail arrival gaps, jump structure)?
3. Include CuFun/UMNN-style monotone flow baselines and MLE Hawkes on synthetic Hawkes. If excluded, justify concretely (code unavailability, compute).
4. Provide operational guidance for ((p, L)): e.g., a heuristic tied to expected event density / hazard smoothness; analyze sensitivity systematically.

---

> ### Author Response · Authors · 2025-11-20
> **Response to Reviewer TZnp (Part 1 of 2, Question 1-4)**
>
> **Q1: In the Related Work you claimed that "TriTPP is only suitable for inhomogeneous Poisson processes". This is incorrect and contradicts the time-rescaling theorem.**
>
> A:
> Thank you for your feedback. We sincerely apologize for failing to clearly state the expressive power of TriTPP in the related work section. I made a mistake while reading the original TriTPP paper. Because I saw a passage in the paper: $$p(\boldsymbol{x}) = | \det J_{\boldsymbol{F}}(\boldsymbol{x}) | \tilde{p}(\boldsymbol{F}(\boldsymbol{x})) = \left( \prod_{i=1}^{N} \frac{\partial}{\partial x_i} f_i(x_1, \dots, x_i) \right) \tilde{p}(\boldsymbol{F}(\boldsymbol{x})).$$
>
> Because the compensator of a Hawkes process does not have a derivative at event times $t_i$ (its left and right derivatives are not equal: $\frac{\partial\Lambda}{\partial t}(t_i^-)\neq \frac{\partial\Lambda}{\partial t}(t_i^+)$), I mistakenly believed that TriTPP could not theoretically represent TPPs whose intensities exhibit jumps, such as Hawkes processes. However, TriTPP is in fact theoretically capable of representing any general TPP, and we have corrected this point in the updated version of our paper. We are very sorry for this lack of rigor.
>
> Nevertheless, we still believe that in practice, TriTPP may not be well-suited for processes such as Hawkes, due to computational efficiency issues and the choice of triangular maps.
>
> - On the one hand, **a triangular map cannot easily capture the recursive Markov structure in Hawkes process intensities**, and can only approximate Hawkes using a triangular map with
> $O(NH)$ complexity.
>
> - On the other hand, **when the triangular map uses RQS monotonic splines, the transformation defaults to a linear interpolation
> $y=x$ outside the transformation window**, assuming that the intensity remains constant over long horizons, rather than decaying exponentially back to the baseline as in Hawkes processes. We believe these issues make TriTPP struggle to fit Hawkes processes well in practice (as shown in both the original paper and our experiments).
>
> In contrast, our MAS model **designs a Transformer as a history encoder and adopts different tail functions** (in addition to the linear tail, we also include exponential tails, etc.), enabling MAS to effectively model various types of TPPs, including Hawkes processes. Thus, we believe MAS has advantages over TriTPP.
> (For more details on how different tail functions adapt to TPPs with different characteristics, please refer to our explanation in Question 6.)
>
> ---
>
> **Q2: There is a typo in the Hawkes CCIF corollary.**
>
> A: Thank you for pointing this out! We have corrected this in the new version and checked the whole manuscript.
>
> ---
>
> **Q3: Regarding the statement that "monotone splines cannot reliably preserve global monotonicity," the concern is that this may be an exaggeration. In flow-based methods, RQS and related spline transforms can remain monotonic when linear tails are used.**
>
> A: Thank you for the clarification. You are correct that our original wording was not sufficiently precise. In the revised version of the manuscript, we now explicitly state that monotone splines composed purely of polynomial pieces and **without any linear tails** may fail to preserve global monotonicity.
>
> ---
>
> **Q4: Some of the performance advantages of MAS fall within one standard deviation. Also, some results show higher variance for MAS than baselines, hinting at stability/tuning issues.**
>
> A: Thank you for the comment. In the experiments, for fairness, MAS was trained using the same set of hyperparameters and the same training configuration across all datasets, without per-dataset tuning. While the unified setting of MAS ensures a fair comparison, it may not be optimal for every dataset, which we suspect contributes to the relatively large standard deviations observed on a small subset of datasets.
>
> However, even under this constraint, we believe the overall performance advantage of MAS remains clear. On most datasets **(Hawkes1, Hawkes2, Renewal1, Renewal2, Earthquake, Taxi, Taobao)**, even when taking three standard deviations into account, **MAS still achieves state-of-the-art results on at least one metric across twelve baselines**. We therefore consider the superiority of MAS to be well supported by the empirical evidence.
>
> ---
>
> **Please see Part 2 for responses to the remaining questions (additional baseline models, tail functions, and hyper-parameter selection guidance).**

---

> > ### Author Response · Authors · 2025-11-20
> > **Response to Reviewer TZnp (Part 2 of 2, Question 5-7)**
> >
> > **Continued from Part 1.**
> >
> > ---
> >
> > **Q5: Please include additional baseline models: contemporary CDF/CCIF approaches (e.g., CuFun / UMNN-style monotone flows) and parametric MLE Hawkes on synthetic data.**
> >
> > A: Thank you for the suggestion. We have added both the CuFun model and the Hawkes model to our experiments. CuFun was evaluated on both univariate and multivariate datasets, and the Hawkes baseline was evaluated on the univariate datasets. However, we were unable to find prior work applying UMNN-style monotone flows to temporal point processes, and thus we have not included this baseline at the moment.
> >
> > ---
> >
> > **Q6: Explain, with evidence, when MAS’s exponential tail outperforms RQS linear tails (e.g., long-tail arrival gaps, jump structure).**
> >
> > A: Thank you for the suggestion.
> > When the inter-event time exceeds the interpolation length, the intensity function becomes fully determined by the extrapolation. In this regime, **if the dataset exhibits Hawkes-like dynamics, i.e., self-excitation followed by gradual decay, an exponential tail is more suitable. In contrast, if the dataset behaves more like a Poisson process, where the intensity tends to stabilize toward a constant value, a linear tail is a better choice.**
> >
> > We also conduct additional ablation experiments in **Table 8** (in the appendix) to examine the role of the tail function further.
> > The results in **Table 8** show that, on the Poisson process, MAS with a linear tail performs slightly better than with an exponential tail, whereas the exponential tail yields better performance on the Hawkes process. Moreover, under the Poisson & exponential-tail configuration, increasing the total interpolation length improves model performance. This is consistent with our theory.
> >
> > ---
> >
> >
> > **Q7: Provide operational guidance for hyperparameters.**
> >
> > A: Thank you for your feedback. At the end of the Generalization Analysis section, we have added a discussion on the selection of hyperparameters. **Specifically, we provide a new corollary, Corollary 4.5 in the revised version**: When the generalization bound is minimized, the total interpolation length $L$ and interpolation interval $\Delta$ satisfy:
> >
> > $$L\ge \sqrt{\frac{\mathbb{E}[t_i] \cdot \lambda_{\text{min}}}{l_1}}, \Delta\le\frac{(Z)^{\frac{1}{3}}(\mathbb E [t_i])^{\frac{5}{6}}\sqrt{\lambda_{\text{min}}}}{\sqrt{l_1}q^{\frac{1}{3}}(24N_0)^{\frac{1}{3}}}.$$
> >
> > where $\mathbb E[t_i]$ presents the expectation of the event interval, $q$ is the freedom of each interpolation knot, $\lambda_{\text{min}}$ is the base intensity during $[0,T]$, and $l_1$ is the Lipschitz constant of the intensity during event intervals.
> >
> >
> >
> > According to **Corollary 4.5**, hyperparameters $L$ and $\Delta$ depend on two dataset-specific quantities:
> >
> > - the expected inter-event time $\mathbb{E}[t_i]$,
> > - the degree of temporal variability in the TPP intensity, captured by its Lipschitz constant $l_1$.
> >
> > As $\mathbb{E}[t_i]$ grows, a longer interpolation horizon, i.e., a larger $L$, is required to track changes in the CCIF accurately.
> > Moreover, as the intensity fluctuation, inflected by $l_1$, becomes sharper, MAS should provide a finer-grained approximation, which requires a smaller interpolation $\Delta$.
> >
> > Furthermore, **Corollary 4.5** provides a simple plug-in strategy for selecting $L$ and $\Delta$:
> > For synthetic datasets where the functional form of the intensity is known, $l_1$, $\lambda_{\min}$, and $\mathbb{E}[t_i]$ can be estimated directly from samples, yielding an approximate lower bound for $L$ and $\Delta$.
> > For general real-world datasets, inspired by Scott’s rule, we suggest assuming that the data follow a Hawkes process and estimating $L$ accordingly.
> > The detailed procedure is provided in the appendix.
> > Given the estimated $L$ and $\Delta$, the number of interpolation knots $p$ can also be calculated as $p=L/\Delta$.
> >
> > It is worth noting that, although an excessively large $L$ and an overly small $\Delta$ may theoretically lead to overfitting and thus increased generalization error, such overfitting can be effectively mitigated by standard training techniques (e.g., dropout [1] or SGD-based optimization [2]). In practice, we observe that this overfitting effect is mild.
> > Therefore, we recommend using a moderately large $L$ and a relatively fine $\Delta$, consistent with the bounds provided in **Corollary 4.5**.
> >
> > **Reference**
> >
> > [1] Nitish Srivastava, et al, Dropout: A Simple Way to Prevent Neural Networks from Overfitting.
> >
> > [2] Nitish Shirish Keskar, et al, Improving Generalization Performance by Switching from Adam to SGD.

---

> ### Author Response · Authors · 2025-11-26
>
> Dear Reviewer TZnp,
>
> If we understand correctly, your main concerns involve: (1) the accuracy of our discussion of TriTPP, (2) several technical clarifications regarding monotone spline monotonicity, typos in the Hawkes corollary, and MAS’s variance, (3) the completeness of baseline comparisons, (4) when exponential vs. linear tails should be used, and (5) guidance on selecting MAS hyperparameters.
>
> In our response, we corrected our original statement about TriTPP, clarified the monotonicity issue, fixed the typo, and explained the variance observations. We also added the requested baselines (CuFun and Hawkes), provided empirical evidence showing when exponential tails outperform linear tails, and included principled hyperparameter guidance through Corollary 4.5 in the revised version.
>
> We kindly ask you to review our replies and let us know whether they address your concerns. Please feel free to let us know if anything remains unclear. We sincerely appreciate your time and constructive feedback.

---

### Author Response · Authors · 2025-12-01
**Response Summary to Reviewer Concerns (Submission8352)**

Dear Area Chair,

Following the recent OpenReview incident and the decision to revert reviews and scores to their pre-discussion state, we would like to briefly document the review history of this submission. We hope this summary assists the new Area Chair and the reviewing team in evaluating the progress made during the discussion period.

This paper introduces Monotone Alternating Spline (MAS), an efficient and flexible parameterization framework for multivariate TPPs. MAS  greatly expands the expressiveness of CCIF-based parameterizations and further improves computational efficiency.
Theoretically, we analyze MAS's approximation and generalization error, and further provide practical guidance for hyperparameter selection.
Additionally, we evaluate MAS against 13 baseline methods on 9 benchmark datasets, where MAS consistently delivers significant improvements.

During the rebuttal stage, we addressed the questions raised by all four reviewers and clarified or revised the identified weaknesses. Specifically:

1. Regarding the concerns raised by **Reviewer TZnp**, we first corrected several inaccuracies in the paper, such as the interpretation of TriTPP and a formula typo. To further demonstrate the superiority of MAS, we included two additional baselines: parametric Hawkes and CuFun. Furthermore, we added an extended ablation study showing that the tail function in MAS can better capture the diverse characteristics of TPP data. Lastly, based on our generalization analysis, we provided detailed and actionable theoretical guidance for selecting the two key MAS hyperparameters — the interpolation interval length and the number of interpolation points.

2. For the concerns raised by **Reviewer SkWr**, we included extensive additional experiments, such as comparisons between MAS and recent SOTA works (HyPro, DTPP), as well as new results for MAS and other baselines on the long-horizon prediction task. We also provided a rigorous theoretical justification for hyperparameter selection, making it no longer purely empirical but truly optimal with respect to generalization error. We further clarified several MAS-related issues, such as how MAS predicts events occurring far into the future and whether interpolation positions are learnable.

3. In response to **Reviewer mUbi**, we reiterated our technical and theoretical contributions: compared with other spline-based or CCIF-based methods, MAS supports direct MLE training, breaks the convexity constraints of MNNs, and eliminates the need for double autograd. MAS offers significant improvements in both performance and computational efficiency, and is backed by comprehensive theoretical guarantees (approximation, generalization, and hyperparameter selection). Additionally, we compared MAS with more commonly used baseline models in terms of efficiency and performed more detailed cross-validation. We also revised several imprecise descriptions in the paper.

4. Concerning the concerns from **Reviewer wrnA**, we improved the overall presentation quality of the manuscript, added clearer and more readable figures, and included more statistical reporting (mean/variance) of experimental results for better rigor.

In summary, our rebuttal provided comprehensive, concrete, and well-supported responses to every concern raised by the reviewers. **Currently, two reviewers have responded to our rebuttal**. **Reviewer wrnA expressed strong positive recognition of our work,** for which we are very grateful. **Reviewer SkWr also confirmed that our rebuttal addressed almost all concerns;** we subsequently provided further detailed clarification regarding the remaining questions about certain baseline model results.

Unfortunately, prior to the score freeze, neither Reviewer TZnp nor Reviewer mUbi had responded to our rebuttal. **Nevertheless, we believe that the evidence and improvements presented in the rebuttal are sufficient to resolve all of their concerns.** During the rebuttal period, we substantially and thoroughly enhanced the paper:

- We provided a more in-depth analysis of the advantages of MAS over prior methods (e.g., MNN, TriTPP).
- We refined the theoretical analysis (including added theoretical hyperparameter selection).
- We introduced a large number of additional experiments, including:
  - four new baseline models,
  - long-horizon prediction tasks,
  - comprehensive cross-validation,
  - more detailed ablation studies,
  - and expanded efficiency tests.
- We rigorously corrected typos and inaccurate or non-standard wording.

Our intention is simply to ensure the new Area Chair is aware that a careful round of rebuttal-driven re-evaluation had already taken place. Although Reviewers SkWr, TZnp, and mUbi did not have the opportunity to update their scores or have not yet replied to our rebuttal, **we have strong reasons to believe that the revisions comprehensively address their concerns and would merit a higher overall evaluation for the paper.**

---

### Meta-Review · Area_Chair_ewdS · 2026-01-13

**Summary:**

This submission proposes Monotone Alternating Splines (MAS) to parameterize the conditional cumulative intensity function (CCIF) for multivariate temporal point processes via a spline-based interpolation region and a simple monotone extrapolation (tail) region. The approach is motivated by improving efficiency and avoiding numerical integration error, and is supported by theoretical analysis and broad experiments on synthetic and real datasets. While the paper is clearly written and the method is practical, the overall contribution is viewed as incremental relative to existing spline/CCIF parameterizations, and the empirical and positioning issues raised by multiple reviewers reduce confidence in the claimed advantages.

**Reviewer Concerns:**

The main concerns are that the proposed method is viewed as an incremental refinement of existing spline- and CCIF-based temporal point process models rather than a clearly novel modeling paradigm. Reviewers also noted initial inaccuracies in positioning prior work and requested stronger empirical validation, including more complete baselines, long-horizon prediction results, and clearer efficiency comparisons. Although many of these issues were addressed in rebuttal, some skepticism remains about the overall significance and generality of the contribution relative to recent alternatives.

**Reviewer Scores:**

Many of the concerns (particularly from the reviewers with lower scores) were more focused on the novelty and incremental nature of this work. I do not think this would have changed much with the rebuttal.

---

### Decision · Program_Chairs · 2026-01-26

Reject